# Expanding biochemical knowledge and illuminating metabolic dark matter with ATLASx

Homa MohammadiPeyhani [1,2,4], Jasmin Hafner [1,3,4], Anastasia Sveshnikova [1], Victor Viterbo[1] &
Vassily Hatzimanikatis [1✉]

Metabolic "dark matter" describes currently unknown metabolic processes, which form a
blind spot in our general understanding of metabolism and slow down the development of
biosynthetic cell factories and naturally derived pharmaceuticals. Mapping the dark matter of
metabolism remains an open challenge that can be addressed globally and systematically by
existing computational solutions. In this work, we use 489 generalized enzymatic reaction
rules to map both known and unknown metabolic processes around a biochemical database
of 1.5 million biological compounds. We predict over 5 million reactions and integrate nearly 2
million naturally and synthetically-derived compounds into the global network of biochemical
knowledge, named ATLASx. ATLASx is available to researchers as a powerful online platform
that supports the prediction and analysis of biochemical pathways and evaluates the bio-
chemical vicinity of molecule classes (https://lcsb-databases.epfl.ch/Atlas2).

[1] Laboratory of Computational Systems Biotechnology, École Polytechnique Fédérale de Lausanne, EPFL, Lausanne, Switzerland. [2] Present address:
Pharmaceutical Research and Early Development, Roche Glycart AG, 8952 Schlieren, Switzerland. [3] Present address: Department of Environmental Chemistry,
EAWAG Swiss Federal Institute of Aquatic Science and Technology, Überlandstrasse 133, CH-8600 Dübendorf, Switzerland. [4] These authors contributed
equally: Homa MohammadiPeyhani, Jasmin Hafner. ✉email: Vassily.Hatzimanikatis@epfl.ch

Metabolic "dark matter" designates biochemical processes where knowledge is still sparse, limiting our general understanding of metabolism, the discovery of key disease mechanisms[1,2], and the development of medicines derived from plant natural products[3]. Metabolic knowledge gaps also hamper the advancement of bioengineering applications like the creation of sustainable cell factories for the green production of commodity chemicals and pharmaceuticals. Key examples of metabolic dark matter include underground metabolism resulting from promiscuous enzymatic activity[4,5], undetected plant natural products and their uncharacterized biosynthesis pathways, and chemical damage of metabolites[6].

Given the vastness of these and other unknown metabolic elements, it is essential to generate hypotheses on potential biochemical functions that guide the experimental discovery of enzymatic functions and natural products. While genomic, transcriptomic, proteomic, and metabolomic data have the potential to generate important hypotheses on metabolic dark matter, linking these data to metabolic functions remains difficult[7,8], and major gaps in biochemical and metabolic knowledge remain. As an example, 25% percent of proteins in *E. coli*, one of the best studied model organisms, do not have a function assigned[9]. In addition, almost 10,000 metabolites in the Kyoto Encyclopedia of Genes and Genomes (KEGG)[10] are considered to be orphan, meaning that they are not integrated in any biochemical reaction within KEGG. An unbiased and global approach is needed to systematically explore the metabolic dark matter arising from the elasticity of enzymatic catalysis, which can be achieved by current computational approaches.

Computational approaches to biological questions have attracted increasing interest over the past few decades. Diverse computational tools have emerged that can bridge knowledge gaps in metabolism through cheminformatic predictions of uncharacterized metabolic reactions, metabolites, and enzyme functions. Most of these tools have been developed for metabolic engineering applications, where the objective is to find biosynthetic routes that produce a desired target compound in a host organism[11–15]. Identification of these biosynthetic routes is accomplished by biochemically "walking back" from the target to precursor metabolites that are produced by, or fed to, the host organism. This procedure is called *retrobiosynthesis* and is implemented in a range of tools such as BNICE.ch[16,17], GEM-Path[18], NovoPathFinder[19], NovoStoic[20], ReactPRED[21], RetroPath[22,23], and Transform-MinER[24]. Retrobiosynthetic methods rely on the concept of *generalized enzymatic reaction rules*. A reaction rule encodes the biochemistry of a substrate-promiscuous enzyme by describing the pattern of the reactive site recognized by the enzyme, as well as the bond rearrangement performed by the enzyme on the substrate. By applying the rule on a substrate that is non-native to the represented enzyme, the rule can predict if (i) the substrate can be recognized by the enzyme, (ii) if the biotransformation can occur, and (iii) the identity of the product molecule(s). While tools featuring reaction rules have the power to predict biochemical reactions, their application is usually limited to a given research or engineering question.

One exception is the ATLAS of Biochemistry database[10], which attempts to map dark matter in biochemistry by predicting reactions between metabolites from the KEGG database[25]. ATLAS contains ~150,000 hypothetical enzymatic reactions predicted by the retrobiosynthesis tool BNICE.ch and annotated with putative enzymes suggested by the enzyme prediction tool BridgIT[26]. In contrast to other tools that use automatic rule generation, the reaction rules in BNICE.ch are designed by experts based on biochemical knowledge and assigned the corresponding three-level Enzyme Commission (EC) number, which

is a numerical coding used to classify enzyme-catalyzed reactions. The complementary tool BridgIT uses the knowledge of the reactive site encoded in the BNICE.ch rules to predict enzymes that can potentially catalyze hypothetical and orphan reactions. More than 100 reactions predicted by BNICE.ch and stored in ATLAS were validated in 2019 following their addition to KEGG, supporting the predictive utility of the ATLAS tool[27]. In addition, Yang et al. were able to experimentally validate predicted ATLAS reactions while designing one-carbon assimilation pathways[28], further demonstrating the value of predictive ATLAS reactions in metabolic engineering. One major drawback of ATLAS is its limitation to KEGG compounds, which excludes many drugs and plant natural products with undefined or putative biological functions. Predicting enzymatic reactions from biochemical compounds retrieved from databases other than KEGG will help expand the scope of our predictions and enhance the application range and the predictive power of the database.

In the following, we present ATLASx, an online biochemical resource providing reliable predictions of biochemical reactions and pathways for synthetic biologists and metabolic engineers. The ATLASx workflow (Fig. 1) unifies biochemical reactions and compounds from 14 different database sources into one curated dataset called bioDB. bioDB holds 1.5 million unique biological or bioactive compounds and 56,000 unique biochemical reactions, which enable the prediction of a hypothetical biochemical space. By applying 489 bidirectional, generalized reaction rules from BNICE.ch onto biological and bioactive compounds within the database, we predicted around 1.6 million potential biotransformations between bioDB compounds. Another 3.6 million reactions were found to connect bioDB compounds with molecules only found in chemical databases, producing a total of 5.2 million predicted reactions. From these predictions, we characterized the connectivity and reactivity of biologically important molecules, and we showed that ATLASx pathway predictions could recover 99% of known biological pathways from MetaCyc. Finally, we provide access to ATLASx through an online web interface that features tools for pathway design and network exploration, which can be accessed at https://lcsb-databases.epfl.ch/Atlas2. The ATLASx platform can be readily used for the design of metabolic pathways, and for the exploration and expansion of biosynthesis pathways. ATLASx distinguishes itself from other tools and web services by providing a database of predicted reactions at an unprecedented scale, annotation quality, and user-friendliness (for a more detailed comparison, see Supplementary Discussion and Supplementary Table 1). In contrast to other prediction tools that dynamically solve one problem at a time, the static nature of predicted reactions stored in a database is an opportunity for experimental scientists to validate the predictions in the short and long-term future. Finally, ATLASx provides an estimation on the staggering number of unknowns in biochemistry and can thus foster future research explorations into metabolic dark matter.

## Results

**bioDB unifies over 1.5 million unique compounds.** Current biochemical databases are heterogenous in their organization, biological scope, and level of detail, which complicates the comparison of data across databases. This in turn makes it difficult to reliably detect biochemical knowledge gaps, as information that is missing in one database may be present in another resource. To reconcile the heterogeneity of existing biochemical data sources, we unified compound data from a variety of biological and bioactive databases (KEGG[25], SEED[29], HMDB[30], MetaCyc[31], MetaNetX[32], DrugBank[33], ChEBI[34], ChEMBL[35]) (Supplementary Table 2). The unification resulted in a reference database, named bioDB, containing 1,500,222 unique 2D structural compound entries.

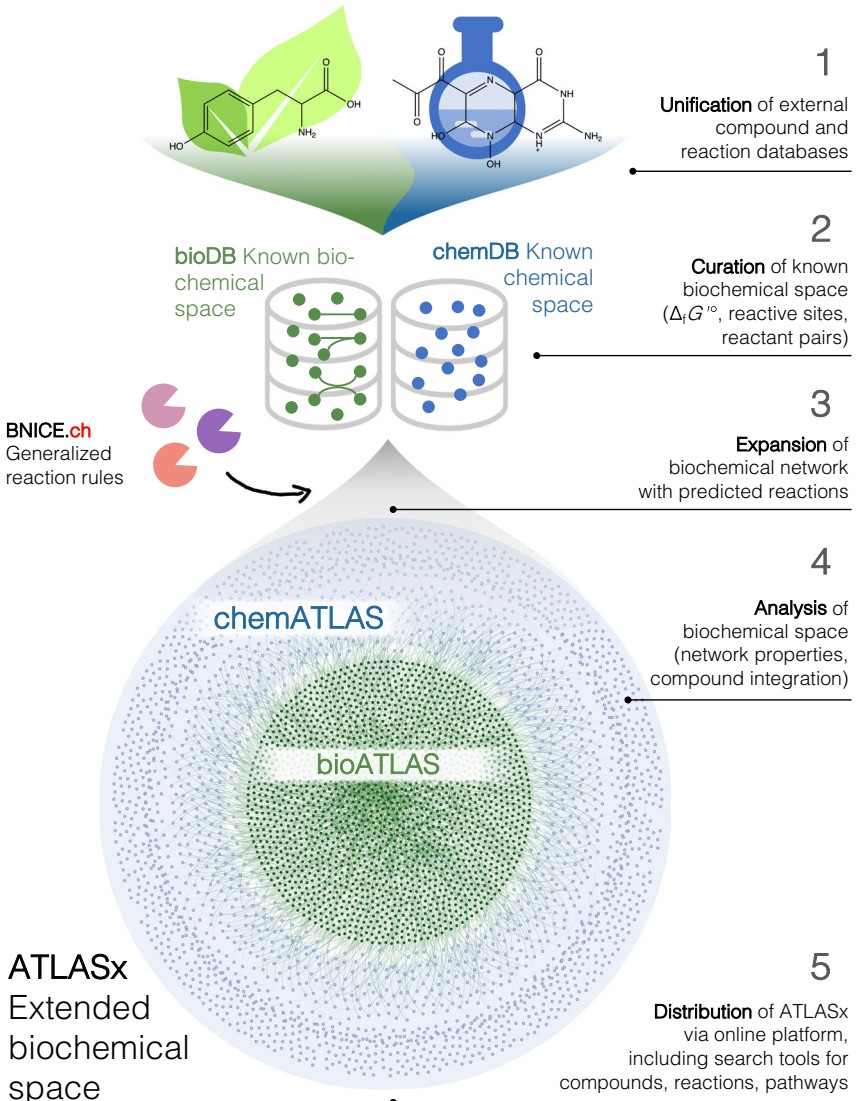

**Fig. 1 ATLAS workflow applied to known biological and bioactive compounds.** 1. Unification- collection of metabolic reactions and biochemical compounds from different publicly available databases, which were merged into a consistent and duplicate-free database, called bioDB. 2. Curation- compounds were annotated with molecular identifiers and reactions were annotated with reaction mechanisms. 3. Expansion- generalized reaction rules from BNICE.ch were applied to bioDB compounds to generate all possible reactions producing known biological or chemical products. 4. Analysis- the connectivity of the biochemical reaction networks was analyzed before and after reaction prediction, as well as the integration of compounds not previously connected in known biochemical networks. 5. Distribution- the results were made available online (https://lcsb-databases.epfl.ch/Atlas2). $\Delta_f G'^o$: estimated Gibbs free energy of formation for compounds under biological conditions.

The contributions of each database to the total of bioDB compounds varied significantly. The three biological databases (KEGG, SEED, and MetaCyc) focus on metabolites and their enzyme-catalyzed interconversion within metabolic pathways and networks and contributed 22,447 (1.5%) of the total compounds within bioDB. The remaining 1,477,775 (98.5%) compounds were contributed by databases covering all compounds produced by, or known to interact with, biological systems (i.e., bioactive compounds). Remarkably, the lion's share of these bioactive compounds (1,447,079, or 97%) came from ChEMBL, suggesting that this database has the most comprehensive definition of bioactivity. The distinction between biological and bioactive compounds reflects the level of biochemical knowledge available for a given compound, i.e., the metabolic interactions of a compound present in both biological and bioactive databases tend to be better characterized than the interactions of a compound only found in a bioactive database. Based on our unification

criteria, we found that the ratio of unique compounds varied across databases. The KEGG COMPOUND database had the highest number of unique compounds (80%, 15,064 unique compounds), while KEGG DRUGS had the lowest ratio (40%, 4514 unique compounds), illustrating the heterogeneity of curation standards between different resources.

To pave the way for integrating chemicals into biochemical pathways, we imported all compounds from the chemical database PubChem[36] (77,934,143 unique molecules). PubChem entries that could not be matched to any existing compound in bioDB were assigned to the chemical compound space (chemDB) in our database, regardless of their true origin (i.e., chemical synthesis, natural biosynthesis, or semisynthetic procedure). Though some of the 77,934,143 unique compounds from the chemical space may still be of biological origin, they may not be labeled as such, or may have the potential to be derived from biological compounds in a bioengineering setting. This artificial

classification of biological and chemical compounds presents an opportunity to re-assign compounds from the chemical space to the biological space by integrating them into hypothetical biochemical reaction.

**56,000 unique biochemical reactions collected from 9 different sources.** To create a unified reaction database as a reference for known metabolic processes, 235,698 reactions entries were collected from KEGG, BRENDA[37], HMR[38], Rhea[39], BiGG models[40], SEED[41], MetaNetX[32], MetaCyc[31], Reactome[42], and BKMS-react[43], and merged into 56,087 unique bioDB entries (Supplementary Table 3). Surprisingly, many databases contained a high number of duplicate reactions. We observed the highest ratio of unique reactions for KEGG (97%), followed by BKMS (95%). Reactome and BiGG had the lowest percentage of unique reactions (49% and 50%, respectively), indicating that many of the reactions are structural duplicates. This quantitative assessment of reaction uniqueness further exposes the heterogenous nature of biochemical databases, suggesting that the number of entries provided by the database hosts should be handled with care when comparing databases, and highlighting the importance of quality assessment. We therefore checked whether or not the collected reactions were elementally balanced and associated with an EC number.

We first searched for reactions containing undefined or un-processable molecular structures (e.g., polymers, proteins, compounds describing two or more disconnected structures such as salts) and other reactions that were not elementally balanced (mostly missing reaction participants, or their reaction mechanism is not known). We found that 56% (31,711 out of 56,087) of total reactions were well-balanced (Supplementary Table 4), and that 46% (25,651) of the reactions had an assigned EC number. The highest ratio of balanced reactions having an EC number assigned was found in the Brenda and KEGG database (80% and 67%), while BiGG had the lowest ratio (21%). The comparatively high number of unbalanced reactions in collections of genome-scale models is partially explained by gap-filling efforts, where hypothetical, unbalanced reactions are added to the metabolic network to ensure model feasibility. This example illustrates how different applications require different levels, scopes and curation standards, resulting in heterogeneous data collections. Our unification and quality assessment of biochemical databases provides an overview on accumulated biochemical data, and it compares curation standards, biochemical coverage, overlaps and consistency across different resources. The resulting unified biochemical space (bioDB) forms the basis for the subsequent expansion of biochemical knowledge through reaction prediction.

**Reactive sites detected in all biological and almost all bioactive database compounds.** Functional groups, or reactive sites, designate which parts of a molecule are recognized and transformed by enzymes, and are important features of biochemically active compounds. To determine the biochemical reactivity of our collected biological and bioactive compounds, we applied 489 reaction rules from BNICE.ch to search for reactive sites within the 1,500,222 biological and bioactive compounds in bioDB, excluding those with more than one disjoint molecular structure (e.g., salts). We found that 1,498,307 out of 1,500,222 (99.8%) of collected biological and biochemical compounds had at least one reactive site (Supplementary Fig. 1a). The remaining 1,915 compounds do not seem to have the biochemical capacity to participate in any enzyme-catalyzed reaction, but their presence in biological databases can still be justified through their interaction with living organisms (e.g., chemically synthesized

molecules with medical or research applications, Supplementary Fig. 1b).

The number and types of reaction rules assigned to a compound is as an indicator for the diversity of functional groups, or the biochemical versatility, of the molecule. By screening our database of biological and bioactive molecules for reactive sites, each compound was assigned a list of reaction rules that can recognize one or more reactive sites on the molecule, thus characterizing its biochemical reactivity (Supplementary Fig. 2). We showed that almost all molecules in bioDB have the potential to undergo biochemical transformations.

**ATLASx predicts 5.2 million hypothetical reactions.** To explore the hypothetical biochemical space and map metabolic dark matter, we used biological and bioactive compounds within bioDB as a seed for reaction prediction. Hypothetical reactions were predicted by applying 489 bidirectional reaction rules from BNICE.ch to the 1,498,307 compounds with at least one reactive site in bioDB. Reactions whose products were part of the assigned biological and bioactive compounds space were stored in the bioATLAS data collection, and reactions for which at least one product was only found in the chemical compound space were stored in chemATLAS. In total, we reconstructed 11,172 of the metabolic reactions in bioDB, and we predicted 5,236,833 hypothetical reactions from biological and bioactive compounds (Table 1). Out of these reactions, 1,590,057 (30%) occurred exclusively between biological and bioactive compounds (bioA-TLAS), and the remaining 3,646,776 reactions involved at least one compound from the chemical space (chemATLAS). The Gibbs free energy of reaction was estimated for 81% of predicted reactions, and all predicted reactions were assigned a third-level EC number as defined by the BNICE.ch reaction rule that generated the reaction, and therefore have defined mechanism. Additional analyses showed no significant difference in the distribution of the estimated Gibbs free energy of reaction between known (bioDB) and predicted (chemATLAS) reactions (Supplementary Fig. 3). Predicted reactions within bioATLAS integrated 56% (844,316 out of 1,500,222) of bioDB compounds. From the remaining 655,906 bioDB compounds, an additional 163,460 were integrated in at least one chemATLAS reaction, showing the importance of integrating chemical compounds into biochemical reaction network prediction. These results are even more important considering that 1,485,324 compounds in bioDB are not involved in any known reaction, and thus orphan. Sixty-seven percent (992,874) of orphan compounds could be integrated into at least one predicted reaction in ATLASx. We further found that 863,000 compounds, originally only present in PubChem, could be integrated into the predicted chemATLAS network, meaning that these molecules are situated only one reaction step away from a known biological or bioactive compound. This ability to

**Table 1 Compound and reaction statistics for bioDB, bioATLAS, and chemATLAS.**

| Category | bioDB | bioATLAS | chemATLAS |
|---|---|---|---|
| Compounds | | | |
| Compounds integrated in reaction | 14,902 | 1,007,776 | 1,870,776 |
| Total number of compounds | 1,500,222 | 1,500,222 | 77,934,143 |
| Reactions | | | |
| Known reactions | 56,087 | 56,087 | 56,087 |
| BNICE.ch-curated known reactions | 11,172 | 11,172 | 11,172 |
| BNICE.ch-predicted reactions | 0 | 1,578,885 | 5,225,661 |
| Total number of reactions | 56,087 | 1,634,972 | 5,281,748 |

link orphan and chemical compounds to known biochemistry highlights the utility of our tool in drug discovery and metabolic research. These compounds are potential candidates for secondary metabolites (e.g., plant natural products), unwanted products of side reactions (e.g., damaged metabolites[44]), or bioactive compounds[45] (e.g., drugs, pesticides) with the capacity to be transformed by enzyme catalysis.

**Network analysis of the biotransformation reveals disjoint components**. The connectivity of a biochemical reaction network can help identify missing biochemical links and serve as an indicator for the comprehensiveness of a knowledge base. According to the chemical law of mass conservation, a network that represents perfect biochemical knowledge would be fully connected—every compound would be connected to every other compound through a suite of biotransformations. To assess the connectivity of our reaction networks, we compared network properties of bioDB, bioATLAS, and chemATLAS using graph theory. Graph theory has been previously used to analyze the properties of biochemical networks, but these analyses were either performed on specific organisms or restricted to single databases[46,47]. Here, we estimated the network properties of known and expanded biochemistry through state-of-the-art graph-theoretical metrics.

We created graph-representations of each network, where nodes represent compounds, and edges represent biotransformations between two compounds (detailed in the "Methods" section). For each of the three networks, we counted the number of connected components (i.e., disjoint graphs or islands) (Fig. 2a), and we found that the total number of components increased with the network expansion from bioDB to bioATLAS to chemATLAS (Supplementary Table 5). However, the number of components relative to the size of the network, represented by the average number compounds per component, decreased from 22.6 in bioDB to 9.0 in bioATLAS, and increased again to 12.2 in chemATLAS. This result suggests that disconnected islands created by integrating bioactive compounds become more interconnected after including chemical compounds. To further characterize the networks, we looked at the size distribution of the different components. We found that all three networks were dominated by one single giant component followed by a large number of secondary components that involved at most 515 compounds (Fig. 2b). While the biggest component in bioDB connected 88% of compounds in the network, this number decreased to 58% in bioATLAS and increased again to 68% in chemATLAS. This result supports our first observation, as it indicates that the integration of chemical compounds into the network bridges compound islands in bioATLAS into a denser network in chemATLAS. Our hypothesis is further confirmed by the diameter metrics. To calculate the diameter of a network, one needs to find all the shortest paths between all the possible combination of nodes in the network. The longest shortest path is called *diameter* of the network, and the average length of shortest paths between any two nodes is called the *average path length*. Here, we found that the diameter decreased with the expansion from bioDB (32 steps) to bioATLAS (27 steps), and increased again in chemATLAS (34-step diameter), suggesting expansion of the network towards uncharacterized chemistry and integration of previously disconnected components. The average path length increased monotonically from 7 in bioDB to 9 in bioATLAS, and to 12 in chemATLAS, reflecting the continuous growth of the main component (i.e., the core of biochemical reaction knowledge) during the network expansion process.

**Searching for biological pathways within ATLASx**. The quest for biochemical pathway alternatives is a key challenge in the

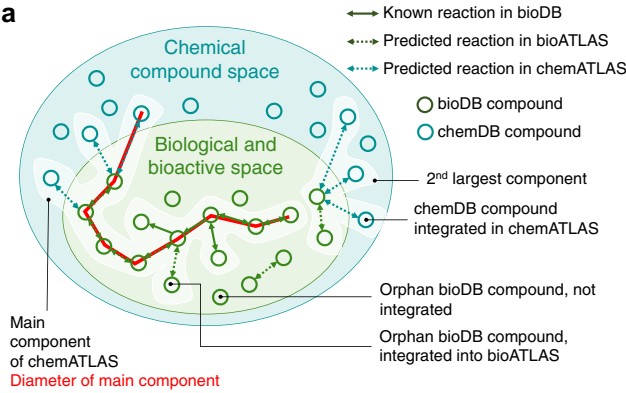

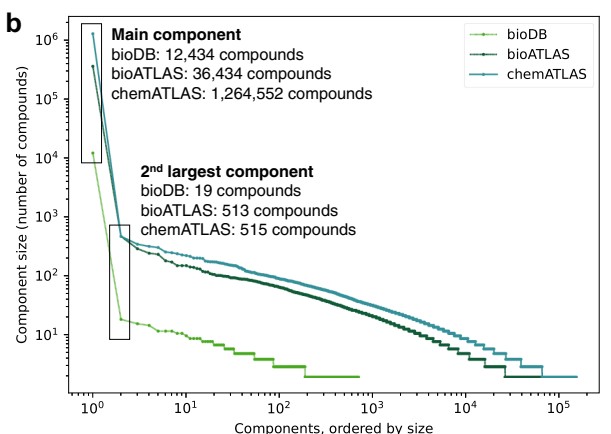

**Fig. 2 Graph-theoretical analysis of biotransformation networks. a** Schematic overview of statistics and network properties calculated for bioDB, bioATLAS, and chemATLAS. Reactions exclusively involving biological and bioactive compounds (green nodes) are assigned to the bioATLAS reaction space, and reactions involving one or more chemical compound (light blue nodes) are assigned to the chemATLAS reaction space. The main component and the second largest component of the network are schematically shown (white highlight). The diameter, or longest shortest path between any two nodes of the main component, has a length of 8 and is highlighted in red. **b** The components of each reaction scope in ATLASx have been extracted and ordered by size. Here, the number of compounds (nodes) of each component is plotted on a log-log scale to show the size distribution of disconnected components for bioDB, bioATLAS, and chemATLAS. For the main component (highest number of nodes) as well as for the second largest component, exact numbers of compounds are indicated.

bioproduction of natural and chemical compounds in chassis organisms, the elucidation of complex natural product biosynthesis, and the study of biodegradation routes. Standard pathway design pipelines include four main steps: (i) creation of the biochemical network; (ii) pathway search; (iii) enzyme assignment; (iv) pathway evaluation in the chassis model[48]. While the steps (i) and (ii) are meant to expand the solution space, steps (iii) and (iv) allow to narrow it down to the number of pathways that can be tested experimentally (top-1, top-5, top-10). Steps (i) and (ii) are automated in ATLASx, and step (iii) can be performed online for a selected pathway by using the integrated BridgIT functionality, making it a valuable tool for the first three steps in pathway design. To support our claim, we validated the ATLASx pathway search on a set of well-characterized biosynthetic pathways obtained from the MetaCyc database.

For this benchmark, we applied the algorithm implemented in NICEpath[49] to search for pathways within the chemATLAS

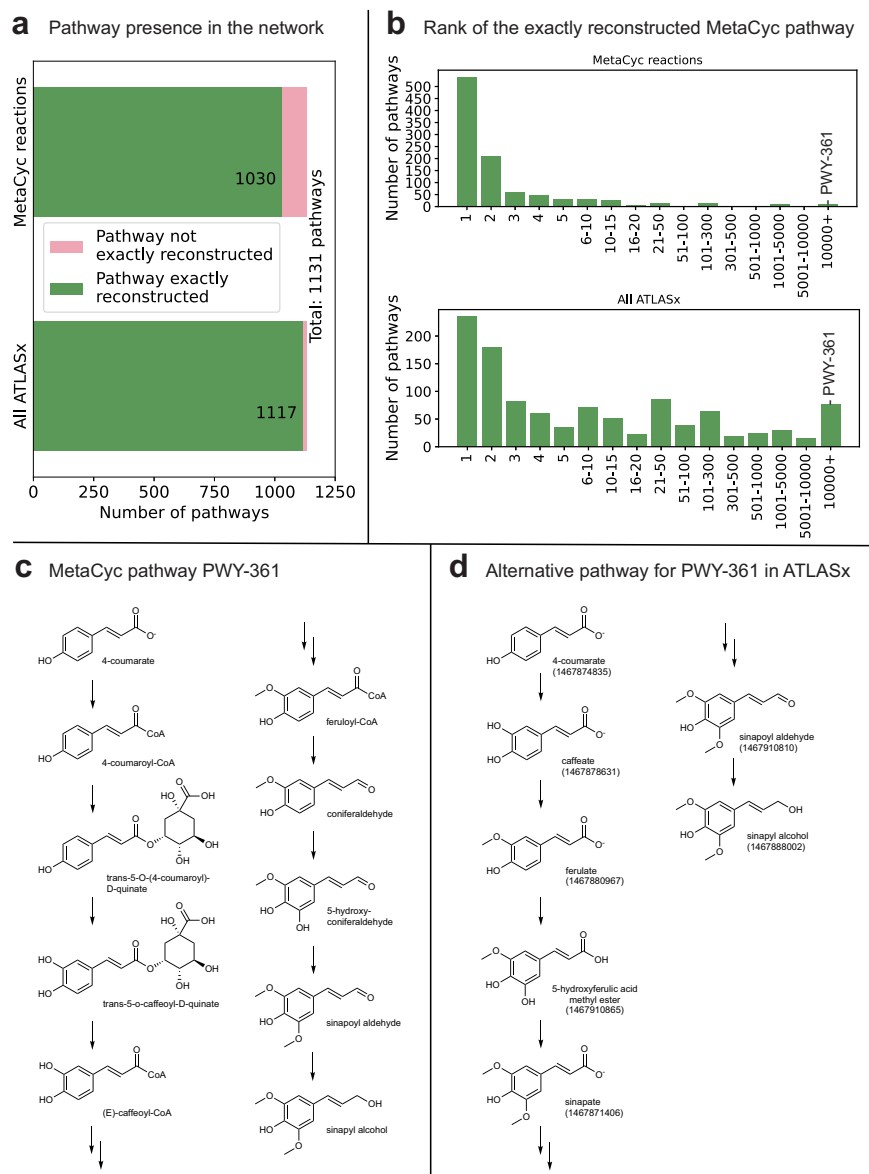

**Fig. 3 Pathway search comparison to dataset of pathways extracted from MetaCyc. a** Coverage of the collected MetaCyc pathways dataset (1131 pathways) with MetaCyc reactions in ATLAS and all ATLASx network. **b** Rank of the MetaCyc pathway according to the NICEpath pathway search algorithm. **c** For the MetaCyc pathway PWY-361 (phenylpropanoid biosynthesis) ATLASx found over 10,000 alternatives with better overall atom conservation. **d** Example of an alternative pathway for the original MetaCy pathway PWY-361. The LCSB IDs of the ATLASx compounds are given within parentheses.

reaction network, and we examined whether ATLASx could recover the pathways from the MetaCyc validation set.

We found that out of the 3149 collected MetaCyc pathways, 1131 matched our curation standards (see "Methods"). For each MetaCyc pathway, we determined the precursor and the target compound. We then used the pathway search within ATLASx to extract biochemical routes connecting the precursor to the target, and we compared the extracted pathways to the original MetaCyc pathway. We were able to find pathways for 99% (1117 out of 1131) of precursor-target pairs within the whole ATLASx (Fig. 3a). We performed the same search within a network containing MetaCyc reactions only, and we found pathways for 91% of precursor-target pairs (1030 out of 1131). Within the MetaCyc network, we further discovered that 85% of native MetaCyc pathways (960 out of 1131) were among the top 15 pathways according to our ranking scheme based on atom conservation (Fig. 3b) (see Methods for details). For the whole ATLAS network, the 15 top-ranked pathways included

the native pathway for the 65% of the MetaCyc pathways (738 out of 1131), demonstrating that the remarkable addition of millions of predicted reactions provided the metabolic community with the tool to obtain pathways with better atom conservation than the currently reported ones while staying efficient in recovering the native pathways. We were further interested to see what are the properties of the pathways that we recover with a low rank. We found, that in these cases the native pathway had steps with low atom conservation, which is possible in natural pathways but is counterproductive for engineering pathways with high yield. For example, sinapyl alcohol biosynthesis from 4-coumarate includes substitution of CoA with L-quinate followed by the reverse substitution after one intermediate reaction step (Fig. 3c). The addition of L-quinate does not bring any additional atoms to the final structure, therefore, our algorithm prefers pathways that are more conservative and therefore we believe are easier to integrate into the host organism and obtain higher yields.

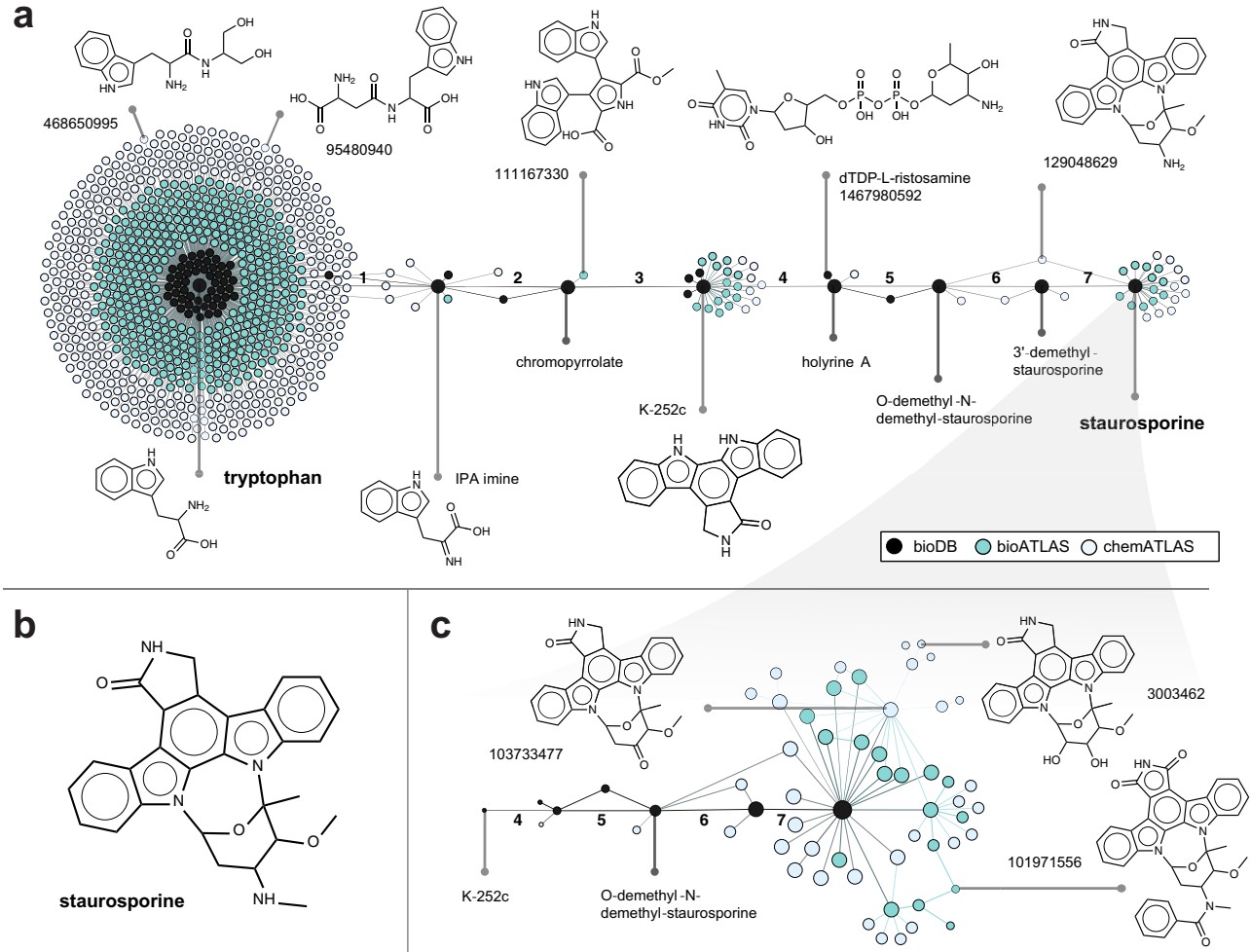

**Fig. 4 Example pathway expansion showcase for the biosynthesis of the natural product staurosporine. a** Visual representation of the biosynthesis pathway from tryptophan to staurosporine (obtained from KEGG, steps numbered in bold black) has been expanded for one generation around the native intermediates. **b** The molecular structure of staurosporine. **c** Network of potential staurosporine biochemistry expanded four generations around the target compound. The size of each compound nodes decreases with each generation. When no compound is indicated, the LCSB compound ID is provided.

Our evaluation shows that known biochemical pathways can be diligently reproduced using ATLASx. The pathway search tool is available online at https://lcsb-databases.epfl.ch/Search2. Users can adjust the network scope as discussed above, as well as perform database-specific search scopes for all of the imported reaction databases.

**ATLASx fills metabolic gaps and predicts biosynthesis pathways.** ATLASx is a resource with wide-ranging practical applications that include compound classification, metabolic pathway searches, and gap-filling for metabolic models. To illustrate how the unification and expansion of knowledge in ATLASx can inform hypotheses around a given biochemical pathway, we used the ATLASx web tools to explore and expand a biosynthetic pathway of interest. As a case study, we chose the biosynthesis pathway of the anti-fungal and anti-hypertensive compound staurosporine, a secondary metabolite with a complex molecular structure (Fig. 4)[50]. According to KEGG, the biosynthesis of staurosporine from tryptophan involves 7 reaction steps, most of them being poorly characterized. To explore the biochemical vicinity of this pathway, we retrieved all compounds that were one step away from the original pathway intermediates. Out of 861 potential pathway derivatives, 60 were found exclusively in bioDB, 407 were contributed by bioATLAS, and the remaining 394 compounds were only integrated when chemATLAS reactions were considered. According to our analysis, most derivatives (93%, or 799) were detected around tryptophan. Secondary hubs were found around K-252c and staurosporine, with each hub contributing 3% (24 compounds) and 3% (22 compounds) to the total number of pathway derivatives, respectively. Intrigued by the high number of potential staurosporine derivatives, we explored the network around this molecule and found 58 derivatives within a distance of four reaction steps (Fig. 4). We found 6 staurosporine derivatives within bioDB (4 of them part of the original pathway), 18 derivatives only within bioATLAS, and an additional 34 compounds within chemATLAS. Interestingly, the network exploration converged, and the only derivatives found four steps away from staurosporine were located upstream of the original pathway.

To characterize the potential staurosporine derivatives we identified, we retrieved the number of patents and citations associated with the compounds in the extracted network, and we ranked the compounds by the sum of patents plus citations. These metrics have been previously established to extract compounds with high industrial, pharmaceutical and academic interest[51]. Among the four top-ranked compounds in the network, we found that staurosporine itself garnered the most attention (29,819 patents plus 15,439 citations), followed by

**Table 2 Pathway reconstruction and gap-filling of the staurosporine biosynthesis pathway with ATLASx.**

| Step | KEGG ID | EC number | BNICE.ch rule | Top BridgIT hit EC (KEGG ID, score) | Reconstruction within ATLASx |
|------|---------|-----------|---------------|-------------------------------------|------------------------------|
| 1 | R11119 | 1.4.3.- | 1.4.3.- | 1.4.3.23 (R09560, 0.95) | Biotransformation with LCSB ID 2600177067 |
| 2 | R11120 | | | | 2-step reaction (spontaneous + 1.21.98.2) in bioDB[a] |
| 3 | R11121 | 1.13.12.- | | | Not reconstructed by any suite of reaction rules |
| 4 | R11122 | 2.4.-.- | | | 3-step reaction in chemATLAS[b] |
| 5 | R11123 | | | | 2-step reaction in bioATLAS[c] |
| 6 | R11129 | | 2.1.1.- | 2.1.2.5 (R03189, 0.34) | Biotransformation with LCSB ID 2600423725 |
| 7 | R05757 | 2.1.1.139[64] | 2.1.1.- | 2.1.1.139 (R05757, 1.00) | Biotransformation with LCSB ID 2600261843 |

[a] https://lcsb-databases.epfl.ch/Graph2/loadPathway/1/1468050408,1469435049,1468050416/2806125367,2806150968/0.
[b] https://lcsb-databases.epfl.ch/Graph2/loadPathway/1/1468050425,1469288899,277921848,1468050433/2603459454,2603467379,2682146339/0.
[c] https://lcsb-databases.epfl.ch/Graph2/loadPathway/1/1468050433,1469288674,1468050440/2603455158,2682148818/0.

7-hydroxystaurosporine (1521 patents, 571 citations) and then K-252c (39 patents, 736 citations), which is part of the staurosporine synthesis pathway. Midostaurin, a cancer therapeutic and protein kinase inhibitor commercially known as Rydapt[52] (158 patents, 570 citations) and was one step away from staurosporine. While "popularity" estimation of chemicals is one of many possible ways to organize and rank compounds, other metrics such as molecular weight, toxicity, or structural similarity to known pharmaceuticals may be applied depending on the objective of the study. This example illustrates how to explore the biochemical vicinity of a compound or pathway within ATLASx, and to retrieve information (e.g., citations and patents) from external sources to rank derivatives based on their academic and industrial relevance.

We further investigated the capability of ATLASx to detect and bridge knowledge gaps in the biosynthesis pathway of staurosporine. Out of the 7 reaction steps in the pathway obtained from KEGG, only one reaction is linked to an enzyme. The other 6 reactions are orphan (i.e., no enzyme assigned), and half of these orphan reactions have unknown reaction mechanisms (Table 2). To show how one can find plausible enzymes(s) for orphan reactions, we examined each orphan reaction within the pathway. First, for reactions with assigned BNICE.ch reaction rules, we applied the computational tool BridgIT to predict potential catalyzing enzymes. For reaction steps without an assigned BNICE.ch rule, we searched for pathways that connect the substrate to the product via alternative sequences of well-annotated bioDB reactions, or hypothetical BNICE.ch reactions that provide the basis for robust enzyme predictions with BridgIT.

The first step of the pathway, the conversion of L-tryptophan to IPA imine, is identified with the partial EC number 1.4.3.- by KEGG. The computational tool BridgIT, which uses enzyme annotations of non-orphan reaction to suggest enzymes catalyzing similar orphan reactions, proposed the enzyme 7-chloro-L-tryptophan oxidase (EC 1.4.3.23) as the best candidate to catalyze this first step. This predication was bolstered by a high BridgIT score of 0.95, which indicates that both substrates have a similar reactive site and surrounding structure. While the native function of this proposed enzyme is to convert 7-chloro-L-tryptophan to 2-imino-3-(7-chloroindol-3-yl) propanoate, the activity of this candidate enzyme on L-tryptophan has been proven in a study by Nishizawa et al.[53]. Another orphan reaction in the staurosporine pathway is the conversion of 3′-demethylstaurosporine to O-demethyl-N-demethyl-staurosporine (step 6). For this reaction, BridgIT suggested that an N-formiminotransferase serves as a catalyzing enzyme (EC 2.1.2.5), although this prediction is accompanied by a relatively low BridgIT score of 0.34. Finally, the last step of the pathway is known to be catalyzed by an O-methyltransferase with EC number 2.1.1.139. In this case, BridgIT successfully mapped this reaction to itself and found the

corresponding enzyme. Interestingly, we also found one reaction step in the pathway that could not be reconstructed by any suite of reaction rules. Although the reaction is known to be catalyzed by an enzyme of the cytochrome P450 class and assigned the partial EC number 1.13.12.-, no information on its reaction mechanism is available from public databases and scientific literature. We therefore hypothesize that the reaction mechanism of this enzyme either involves intermediate molecules that have not been characterized yet, or it harbors reaction mechanism that has not been observed before in nature.

This showcase exemplifies how BridgIT can be used on top of the BNICE.ch reaction prediction to find enzymes for hypothetical or orphan reactions and to fill gaps in metabolic pathways and networks. All of the presented analyses can be performed using the computational tools available online, in combination with open-source visualization software as detailed in the Methods section. We provide public access to our database through an online search interface, which includes a powerful pathway search algorithm that can be used for the design of metabolic pathways. The web access to ATLASx (https://lcsb-databases.epfl.ch/Atlas2) provides further query tools, such as the ability to identify all reactions associated to a query compound.

## Discussion

This work attempts to map the hypothetical vicinity of known biochemistry and to address the vast amount of metabolic "dark matter" by using biochemical reaction principles implemented in 489 generalized reaction rules. Based on 1.5 million known biological and bioactive compounds unified in bioDB, we predicted 1.6 million biochemically possible biotransformations between biological and bioactive compounds (bioATLAS). We further predicted more than 3.6 million reactions that involved compounds from the chemical compound space, resulting in a total of almost 5.2 million in chemATLAS. From this wealth of information, we extracted insightful numbers on the reactivity and connectivity of biologically relevant molecules.

Assessing the composition of metabolic "dark matter" is by definition difficult, since we lack a way to quantify the unknowns a priori. Fortunately, biochemical data collected and generated from our database allows us to answer a broad range of questions regarding the biochemical reactivity of compounds, the expansion of biochemical space from a graph-theoretical perspective, and the characteristics of our hypothetical reaction network. Potential applications of ATLASx include the prediction of bioproduction or biodegradation pathways involved in the transformation of commodity and specialty chemicals, pharmaceuticals, and plastics. ATLASx can also be used to discover the biosynthesis routes of poorly characterized secondary metabolites, and systematically fill in knowledge gaps surrounding metabolic models. For example, ATLASx can be used to expand the network around all compounds within a given metabolic model, remove dead-end

metabolites, and then examine the expanded model for potential shortcuts, enhanced predictions, and enzymatic promiscuity.

Since we successfully integrated tens of thousands of chemical compounds into a biochemical network, we hypothesize that many compounds are not yet part of any database, even though they potentially exist in nature or could be created by metabolic engineering. While the integration and accurate prediction of hypothetical compound structures remains an open challenge, ATLASx provides the necessary tools and conceptual framework to predict hypothetical compounds reliably in the future. In order to properly meet that future, ATLASx is designed as a dynamic database, and can be continuously expanded around biochemical pathways or compound classes of interest. We believe that predictive biochemistry is crucial for the advancement of synthetic biology and metabolic engineering, and hope that ATLASx can provide reliable reaction and pathway predictions for the scientific community.

## Methods

We developed a unification algorithm based on canonical SMILES to generate consistent and duplicate-free data. Data was collected from KEGG[25], SEED[29], HMDB[30], MetaCyc[31], MetaNetX[32], DrugBank[33], ChEBI[34], ChEMBL[35], BRENDA[37], Rhea[39], BiGG models[40], Reactome[42], and BKMS-react[43] by January 2019. To efficiently store, retrieve, and analyze the increased amount of data, we created an SQL-based database where we imported the collected data (bioDB). The compounds were filtered to keep only those that had a defined, single molecular structure (i.e., no salts, no polymers, no generic molecules with undefined branches). Compounds were merged using their canonical SMILES as a unique identifier, which was calculated using OpenBabel[54] version 2.4.0. As a result of the unification procedure, a unique compound entry in bioDB can contain different resonance forms, stereoisomers, as well as dissociated and charged states of a same compound. Since ATLASx does not distinguish between stereoisomers, users who wish to include stereochemical considerations are encouraged to post-process ATLASx output with external cheminformatic software (e.g., rdkit.Chem.EnumerateStereoisomers module in RDKit) to expand the 2D structures to all possible 3D stereoisomers structures.

Reactions were unified based on the structure of their reactants and products. Transport and stereoisomeric reactions were filtered to only keep reactions that modify the connectivity of atoms within the molecule. Unified compounds and reactions were annotated with all available identifiers from different databases

**Curation of unified compounds and reactions**. Compounds were annotated with the following structural descriptors: chemical formula, molecular weight, InChI, InChIKey. For the annotation we used OpenBabel[54] version 2.4.0. Reactions were annotated with enzymatic reaction mechanisms obtained from BNICE.ch. We further estimated the standard Gibbs free energy of the reactions in cellular conditions using the Group Contribution Method (GCM)[55].

**Reactive site analysis**. We used BNICE.ch to determine the reactivity of compounds. For this, we applied all of the 489 bidirectional generalized reaction rules available in BNICE.ch to the unified set of compounds. The collection of BNICE.ch reaction rules has been continuously expanded in the past to account for a wider range of biochemical reaction mechanisms (Supplementary Table 6). Since 2018, the creation of rules was particularly focused on less common reaction mechanisms from secondary plant metabolism and biodegradation of organic pollutants. The BNICE.ch rules rely on a bond-electron matrix (BEM) representation of the reactive site that will be recognized by the enzyme. A second matrix (difference BEM) describes the bonds that need to be rearranged in the molecule to form the product. To determine the potential reactivity of the compounds, we therefore used the first BEM describing the reactive site of each reaction rule to screen all of the molecular structures of the compounds for potential reactive sites. Each compound was assigned all the rules that harbor a matching reactive site for further usage.

For compounds with aromatic rings, all possible kekulé representations were generated to ensure that no potential reactive site is missed. Since the double bonds in aromatic compounds can be drawn in different positions, giving rise to different mesomers, it is important to screen all the different kekulé structures for reactive sites. It has been shown that different kekulé structures lead to different results in cheminformatics approaches like docking, molecular-fingerprint construction, clustering, and any 2D- or 3D-QSAR analysis[56].

**Prediction of hypothetical reactions**. Each BNICE.ch reaction rule was applied to all compounds recognized by the rule in the previous step, and all the generated products were analyzed. Reactions only producing compounds that belong to the biological or biochemical compound space were imported to the database as bioATLAS reactions, and reactions involving products from the chemical space

were imported as chemATLAS reactions. Importing a generated reaction into the database involved checking its equation against all the reactions present in the database. If a reaction was already present, only the reaction rule was added to the reaction description, otherwise the reaction was imported. For each generated reaction, the GCM[55], as integrated in BNICE.ch, was employed to provide an estimation of Gibbs free energy of reaction and its error.

**Network analysis**. Most available methods for network analysis are based on manually derived reactant-product pairs (e.g., KEGG RPAIR network[57]), or they define a set of cofactors to be excluded from the analysis to avoid the generation of hubs by currency metabolites. Here, we rely on NICEpath, a pathway search tool proposing a graph representation of metabolic reactions that weighs each substrate-product pair according the number of atoms conserved between the substrate and the product[49]. NICEpath has been previously shown to be able to retrieve relevant pathways from biochemical networks, and to help in understanding the inherent properties of biochemical networks (e.g., connectivity, diameter).

To calculate the weights on each pair, NICEpath requires each reaction to be annotated with a reaction mechanism that allows the calculation of the atom conservation between the substrate and the product. This condition is met for all bioDB reactions with assigned reaction rules, and for all predicted reactions in bioATLAS and chemATLAS. For reactions in bioDB without reaction mechanism assignment, the atom conservation was estimated by assuming a minimal rearrangement of atoms in the course of the reaction.

For each pair, the conserved atom ratio (CAR) is calculated with respect to the reactant ($CAR_r$) and with respect to the product ($CAR_p$), where $n_c$ is the number of conserved atoms between the reactant and the product, $n_r$ is the number of atoms in the reactant, and $n_p$ is the number of atoms in the product. Hydrogen atoms are excluded from the calculation.

$$CAR_r = \frac{n_c}{n_r}, CAR_p = \frac{n_c}{n_p} \qquad (1)$$

To calculate a bidirectional CAR, the mean $CAR_{r,p}$ is multiplied with a correction factor that increases with the difference between the number of common atoms and the total number of atoms in the molecule.

$$CAR = \frac{CAR_r + CAR_p}{2} \qquad (2)$$

For bioDB reactions that could not be annotated with a BNICE.ch reaction mechanism, $n_c$ was estimated as follows: First, the number of conserved atoms was assumed to be defined for standard cofactors (e.g., $NAD^+$ and NADH conserve 100% of atoms, hydrogen excluded). If no pre-defined cofactor pair could be identified, we estimated the number of common atoms using the maximum common substructure algorithm (implemented as FMCS library in RDKit). For the remaining pairs, the number of conserved atoms was calculated based on the assumption that the maximum possible number of atoms is conserved between a given substrate-product pair (e.g., $C_6O$ is conserved between $C_7O$ and $C_6O_2$, hydrogens excluded). For the remaining reactant pairs, the atom conservation was assumed to be equal to 0.

To create a weighted, searchable biochemical network, the CAR of a substrate-product pair is then transformed into a distance between the substrate node and the product node, and the two are connected by an edge representing the biotransformation.

$$Distance = \frac{1}{CAR} \qquad (3)$$

The result is a reactant pair network where substrates and products that conserve more atoms lie closer to each other than reactants that conserve less atoms. Since a same biotransformation, or substrate-product pair, can occur in more than one reaction, each edge in the network represents all the reactions that transform the same substrate to the same product. The resulting atom-weighted biochemical networks can be searched for metabolic pathways using common graph-search algorithms. For example, NICEpath uses Yen's $k$-shortest loop-less path search algorithm to retrieve the top $k$ pathways with the highest atom conservation from the source to the target compound[58]. Here, we analyze the atom-weighted networks to derive global properties of known and predicted biochemistry.

We constructed three networks with different biochemical scopes: The 56,087 reactions in bioDB translated into 62,299 weighted edges connecting 14,914 bioDB compounds. The bioATLAS network connects 844,337 compounds in 1,590,057 reactions, represented by 2,503,627 edges in the network, and chemATLAS connects 1,876,992 compounds in 5,248,711 reactions, represented by 5,717,409 edges (Supplementary Table 2). For many types of network analysis, however, an unweighted graph is required. Since we know that substrate-product pairs with a very low degree of atom conservation may not be biologically relevant, we used a cutoff CAR value to decide whether or not to draw an edge between any given compounds. It has been shown previously that a cutoff of 0.34 in the CAR best predicts the manually curated reactant pairs of type "main" in KEGG[49,57]. Hence, we removed edges with a CAR below 0.34, and we removed the weights from the remaining edges. The result is a set of unweighted graphs, which can be analyzed using standard graph analysis algorithms (i.e., extraction of disjoint components, diameter calculations on disjoint components). The code used to perform the

network analysis and to create the figures in the manuscript is available at https://github.com/EPFL-LCSB/ATLASxAnalyses, including documentation to reproduce the presented results[59].

**Reconstruction of known reactions with BNICE.ch reaction rules**. To estimate the biochemical coverage of BNICE.ch reaction rules, we assessed how many bioDB reactions could be reconstructed by BNICE.ch rules. For this, we first removed reactions missing structural information on their reactants, as well as isomerase and transport reactions. For the remaining reactions, we checked if the reaction could be reconstructed by one or several BNICE.ch reaction rules. We distinguished between exact reconstruction, where reactants and products of the reaction match exactly, reconstruction of the main biotransformation using alternative cofactors, and reconstruction of the main biotransformation using 2 to 4 reaction steps with BNICE.ch rule within ATLASx. For multi-step reconstruction of reactions, we considered all main reactant pairs (i.e., with a CAR ≥ 0.34) of a given reaction, and we searched to connect the substrate of the pair to the product. To do this, we extracted the shortest path within the weighted reaction network of chemATLAS, considering only BNICE.ch curated reactions. If the substrate and the product could be connected in k reaction steps, and the product of the CARs along the path did not drop below the threshold of 0.34, we called the reactant pair *reconstructed in k steps*. If the reaction could be split into several "main" reactant pairs, the pair reconstruction with the highest number of steps was considered for the reaction reconstruction. Using this workflow, we found that 71.4% of filtered reactions in bioDB could be reconstructed by BNICE.ch (Supplementary Table 7). Considering that only 35% of bioDB have a high curation standard (i.e., mass balanced reactions with EC annotation) (Supplementary Table 4), we consider our coverage of known reactions as sufficient, and we show that we can propose reaction mechanisms for known reactions for which the reaction mechanism is still unknown.

**Pathway reconstruction of linear pathways from MetaCyc**. MetaCyc pathways and reactions were downloaded from MetaCyc (https://metacyc.org/smarttables) on May 25, 2020. The pathways used for benchmarking had to pass the following criteria: The pathway (i) consists of a minimum of two reactions, (ii) it does not contain transport reactions and electron-transfer reactions, (iii) it does not contain reactions that are not listed in the MetaCyc reactions table, (iv) it does not contain compounds with undefined structure, non-carbon compounds, proteins and peptide polymers, RNA molecules, unknown compounds, (v) it is not circular (as we are comparing linear pathways), (vi) is not a polymerization pathway (e.g. bacterial peptidoglycan polymerization), (vii) is not a light-dependent pathway, (viii) is not a superpathway (i.e., pathway consisting of other pathways with no individual unique reaction sequence).

The pathways in MetaCyc are reported as a set of reactions and not as a sequence of compounds. We therefore translated the pathway of reactions into a pathway of compounds to directly compare the output. For these, we generated a graph based on the reaction equations: all the compounds of the reactions were considered as nodes, and biotransformations were introduced into the graph as edges for every substrate-product pair with the same distance. This is because we do not have information about the atom conservation from MetaCyc's sources. At the same time, we excluded the common cofactors (NAD, ATP, etc.) from the pathway graph to minimize the possible number of shortcuts. We found the shortest loop-less linear path between every pair of compounds in this graph, and we considered the longest of these pathways (therefore the 2 more distant compounds) as the linear pathway of reference (e.g. longest branch of the branching pathway). The resulting dataset was further manually checked to exclude the pathways that were incorrectly extracted in the automatic procedure described above (Supplementary Table 8). The resulting linear set of pathways was considered as a reference pathway to benchmark the pathway search algorithm. For the pathway search we applied the following parameters: as the pathway search scope we used four different networks, one involving all ATLASx reactions, one only including reactions that have a BNICE.ch reaction mechanism assigned, only involving all MetaCyc reactions and one including only MetaCyc reactions that have a BNICE.ch mechanism. The analysis of respective differences for networks with and without BNICE.ch mechanism annotation for all reactions is provided in the main text. Edges with a CAR value below 0.34 were removed from the network, and the distance between nodes was calculated by Eq. (3).

We first checked whether all the edges of each MetaCyc path are present in the ATLASx network and therefore the native pathway can be found in ATLASx. Then we applied the NICEpath pathway search algorithm, which finds the shortest pathways according to the distance calculated based on CAR[49]. We searched the ATLASx network for the pathways between the precursor and target until the native MetaCyc pathway was encountered. The rank of the pathway within the network was reported. We also investigated how the length of the reference pathway from MetaCyc affects the proportion of reconstructed pathways. We found that even pathways as long as 16 reaction steps could be exactly reconstructed from the original MetaCyc pathway, and that alternative pathways for MetaCyc pathways could be up to 26 reaction steps in length (Supplementary Figs. 4 and 5). These results show that the performance of the pathway search is not significantly compromised when searching for longer pathways. The code used

to perform the pathway reconstruction and to create the figures in the manuscript is available at https://github.com/EPFL-LCSB/ATLASxAnalyses, including documentation to reproduce the presented results.

**Pathway expansion**. The biosynthesis pathway of staurosporine was obtained from KEGG (https://www.kegg.jp/kegg-bin/show_pathway?map=map00404&show_description=show, 12 May 2020). For each intermediate, the LCSB ID was obtained using the online compound search interface, and the IDs were compiled into a compound list. The ID list was then used as input to extract all compounds one reaction step away from the native pathway for bioDB, bioATLAS, and chemATLAS scopes (https://lcsb-databases.epfl.ch/Atlas2/Analysis, Analysis 1). The default values were applied (CAR threshold of 0.34, reactions without known BNICE.ch reaction mechanism included). The resulting network file for chemATLAS was imported to the open-source graph visualization software Gephi (.csv file extension required for import). To assign the origin to each compound in the network, the list of nodes was obtained from the bioDB and bioATLAS networks, and each node in the chemATLAS network was assigned the corresponding scope. The node table was imported into the Gephi project, and nodes were colored in accordance to their scope. Finally, Gephi's forceAtlas2 algorithm was applied iteratively to the network and the visualization was manually improved. This analysis can be reproduced by following the user guide available on our website (https://lcsb-databases.epfl.ch/pathways/downloads/ATLASx/userguide.pdf).

To assess the "popularity" of the staurosporine derivatives, the number of publications was derived from PubChem and PubMed, while the number of patent annotations was extracted from PubChem. The PUG-REST service was used to retrieve information from the PubChem website (https://pubchem.ncbi.nlm.nih.gov/) on the number of patents and citations associated with each compound. We further used the Entrez Programming Utilities (E-utilities) API service to search the PubMed database for citations by compound name. To find enzymes for the predicted reaction, the BridgIT tool was used[26]. BridgIT is available as a resource on our website (https://lcsb-databases.epfl.ch/Bridgit).

**Website**. The ATLASx search and analysis tools can be found at https://lcsb-databases.epfl.ch/Atlas2. The visualization of compounds is achieved by SMILES viewer, a light-weight JavaScript library developed by Probst and Reymond[60] that visualizes molecular structures in runtime. The pathway search within known and predicted biochemical network uses the concepts and code developed in NICEpath[49]. The Python library NetworkX is used for the pathway search implementation, and the network statistics were obtained using the Python libraries NetworkX and SNAP[61].

**Reporting summary**. Further information on research design is available in the Nature Research Reporting Summary linked to this article.

## Data availability
Data supporting the findings in this work are available within the paper and its Supplementary Information files. Additional source data that are necessary to reproduce the analyses and figures presented in this paper are available on the publicly available git repository (https://github.com/EPFL-LCSB/ATLASxAnalyses). The data stored within the ATLASx database are available from the authors upon reasonable request. The following previously published datasets were used in this work: PubChem compound data (released in 2020, pubchem.ncbi.nlm.nih.gov)[56], KEGG database (released in 2018, www.genome.jp/kegg/)[25], ChEMBL database (released in 2020, www.ebi.ac.uk/chembl/)[35], MetaCyc (released in 2020, metacyc.org/)[31], Model SEED (released in 2020, modelseed.org/)[62], Drugbank (version5.1.6, go.drugbank.com/)[33], ChEBI (released in 2020, www.ebi.ac.uk/chebi/)[34], HMDB (Release 4.0, hmdb.ca/)[30], MetaNetX (version 4.1, www.metanetx.org/)[32], HMR (version 1.6, metabolicatlas.org/)[38], Reactome (released in 2020, reactome.org/)[42], Rhea (release 115, www.rhea-db.org/)[39], BKMS (released in 2019, bkms.brenda-enzymes.org/)[43], BiGG models (version 1.6, bigg.ucsd.edu/)[40], and Brenda (released in 2019, www.brenda-enzymes.org/)[37].

## Code availability
The tools used to build, annotate and search ATLASx have been previously published as BNICE.ch (version 2020)[16], BridgIT (version 2022)[26] and NICEpath (version 2021, https://github.com/EPFL-LCSB/nicepath)[49], respectively. BNICE.ch and BridgIT use the OpenBabel library (version 2.4.0)[54] for structural format conversion. NICEpath uses the NetworkX library (version 2.5)[63] to represent and search biochemical networks. The code to reproduce the presented analyses and figures is available at https://github.com/EPFL-LCSB/ATLASxAnalyses (https://doi.org/10.5281/zenodo.5925282). The python libraries NetworkX (version 2.5) and the SNAP (version 5.0.0)[61] were used for the network analysis.

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

## Acknowledgements

We thank Dr. N. Hadadi for her creative inputs in an early stage of this work. We further thank Dr. K. Butler for valuable feedback in the preparation of the manuscript. Funding for this work was provided by the Swiss National Science Foundation (SNSF) (NCCR Microbiomes grant agreement 51NF40_180575 and MicroScapeX grant number 2013/158 project number 200021_188623), the European Union's Horizon 2020 research and innovation program (Marie Skłodowska-Curie grant agreement No. 72228 and Shiki-Factory grant agreement No 814408), and the Ecole Polytechnique Fédérale de Lausanne (EPFL).

## Author contributions

H.M., J.H., and V.H. designed the study. H.M., J.H., A.S., and V.V. developed algorithms, performed the analyses, and wrote the manuscript. H.M. and A.S. curated the data. H.M. developed the database architecture. J.H. developed the web interface. V.H. was responsible for project administration and funding acquisition, and he conceptualized and supervised the project.

## Competing interests

The authors declare no competing interests.
