## [Peer Review File · Nature Communications]

Reviewers' Comments:

Reviewer #1:

Remarks to the Author:

Hatzimanikatis et al. applied 490 generalized enzymatic reaction rules on 1.5 million biological and bioactive structures to construct a global network of biochemical knowledge map (ATLASx). The known biochemical pathways could be diligently reproduced using ATLASx, however, the capacity of exploring new biosynthetic pathway and unknown biochemical reaction is not well estimated in this work. I agree with the author's view that the exploration of metabolic dark matter arising from the elasticity of enzymatic catalysis is critically important, but I do think that this challenging issue "can be achieved by current computational approaches". I would like to think the ATLAS as a well-established database but not a good predictor after reading the whole paper and visiting their webserver.

My Major Concerns:

1. How can prove the capacity of exploring new biochemical transformation by rule-based method? Are the 490 so-called generalized enzymatic reaction rules sufficient for describing so complicated biocatalysis? They claim that a total of 5.2 million novel reactions were generated by applying the reaction rules. This number is far more than the "valid" reactions (56000 in this work), and it really attracts me if they are truly new. While my queries are, what kind of reaction means novel? how reliable for these hypothetical reactions? is there a score function(not the Gibbs free energy but something like weighted score reflecting the matching between structure and reaction rules) to evaluate the possibility of these reactions? This work is lack of the validation on the virtual bioreaction network, thus the ability of biochemical reaction exploration is not well verified. In sum, they should carefully prove the reliability of computationally constructed map for the hypothetical vicinity of known biochemistry.

2. According to the main text, when applying the reaction rules to the structures, only the reactions whose products exist in bioDB or chemDB were kept. Will novel structures be produced in this process? If yes, I think the new structures are also important to expand the chemical space of biological reactions. So additional database containing the reactions with "virtual" components could be constructed to supplement bioATLAS and chemATLAS.

3. I'm confused about the classification of biological and chemical compounds (i.e. the bioDB and chemDB). Since this work aims to explore the metabolic "dark matter", why not just classified compounds as biological origin (or metabolites) and non-biological origin (including those unlabeled from PubChem and other data sources). For examples as summarized in the supplementary Table S1, I do not understand why put bioactive structures into bioDB. Why put the metabolic data from MetaNetX into bioactive subset of BioDB? There are too many details for the construction of database and reaction network in the main text, this manuscript is suitable for professional reader while not general and friendly for diverse readership.

4. For the webserver, users could not use the web tool listed in the manuscript, if without registration. And the registration for academic use seems cumbersome that too many personal and organization information are required to finally get the License Agreement. In my experience, the software or database are free for academic use and the interface are very friendly for many computational paper published in nature serial journal. While it is so hard for result reproduction by visiting their webserver.

Reviewer #2:

Remarks to the Author:

In this paper, MohammadiPeyani et al. expend our current knowledge of the metabolic "dark matter," the set of possible reactions extrapolated from the known metabolome using a set of generalized enzymatic reaction rules. The authors first generate a unified biochemical database and a chemical one. Using a previous list of 490 generalized enzymatic reactions rules obtained from their BNICE tool, they generate a new set of hypothetical biochemical reactions. While the methodology is similar to their previous works (Hadadi et al. 2016, Hafner et al. 2020), the novelty comes from the starting compound database used. In this study, >5M novel reactions could be inferred versus 150'000 from the previous version (Hafner et al. 2020). ATLASx will be a

much-needed tool to find novel pathways for metabolic engineering and should be of interest to the readership of Nature Communications. The study is well-written with descriptive figures, but additional information could be added to the manuscript to ensure reproducibility and clarity.

Specific points that could be addressed to improve the manuscript:

Regarding the accessibility of ATLASx, I would like to congratulate the authors for making ATLASx available through an online platform. From a user perspective however, accessibility to the database needs to be applied for and approved through the principal investigator. Since the database would probably be mostly used by PhD students and postdocs, I would suggest giving direct access to all researchers regardless of their academic position.

Central to this study are the 490 reaction rules used from BNICE. However, details regarding these rules were absent from the main and supplementary text. Were they derived from KEGG as in the previous studies? Looking at the KEGG database, the current release contains $\approx 12'000$ reactions ($\approx 11'000$ reactions in 2018), is the $>20\%$ increase in reaction rules (400 in 2018 vs 490 now) only due to the addition of 1'000 new reactions?

p.7 The authors mentioned different curation standards across the different databases used. Does the unified version contain all reactions from these databases, others than the duplicate reactions that got removed? For instance, were unbalanced reactions kept in the analysis? Can the authors comment on whether filtering criteria were applied here and the rationale behind the inclusion or occlusion of those?

p. 11, l. 3 The top 100 pathways relative to what metric?

p. 11 "Out of the reconstructed pathways, 40% were exactly reconstructed within the top 100 pathways" This sounds low but again what is meant by "top 100" pathways?

Minor comments:

p. 4 "Yang et al. were able to experimentally validate predicted ATLAS reactions" reference missing in text

ref. 57 authors name in capital letters

Reviewer #3:

Remarks to the Author:

Authors implemented BNICE.ch to construct a biochemical knowledgebase, ATLASx, which guides the analysis of unknown metabolic processes. Combined with the enzyme prediction tool, BridgIT, it is expected that the predicted biosynthetic pathways will help developing microbial cell factories more efficiently. The followings need to be considered to improve the manuscript.

Page 5: The authors implemented 490 bidirectional reaction rules from BNICE.ch. However, unidirectional reactions are also prevalent and important for the construction of reaction networks. If applicable, please add unidirectional reaction rules.

Page 5: bioDB unified 1.5 million unique compounds in 2D structural compound entries. Can the entries show differences between isomers? Does the ATLASx contain compounds with various isomers and corresponding reactions?

Page 6: The compiled compounds drugs (or drug-like compounds) broaden the utility of the database for drug discovery. Then, it would be beneficial to provide molecular property distribution of the compiled compounds (e.g., partition coefficient, quantitative estimate of drug-likeness).

Page 8: Almost all molecules in bioDB were applicable for the BNICE.ch reaction rules. Was it due to the specificity of the reaction rules for molecules in bioDB? Can molecules not in the bioDB have

the potential to undergo biochemical transformations?

Page 10: Please elaborate on the general description of the NICEpath (how it works, what does the ranking means.) in the methods section.

Page 11: The authors searched the top 100 pathways to validate the pathway search algorithm. However, 100 predictions are still too much to examine in wet experiments. Performance evaluation of general retrosynthesis algorithms analyze with top-1, top-5, top-10 searches. Perform the analysis with various thresholds.

Page 11: The authors showed 99% of the precursor-target pairs were found. As the former analysis performed the predictions with the threshold of top 100 pathways, the value would be better to be 46% (which was the number of exactly reconstructed pathways within the top 100 pathways). Otherwise, it might confuse readers.

Page 13: The investigation of the biosynthesis pathway of staurosporine is the application of the previously developed algorithms (BNICE.ch and BridgIT), rather than the novel analysis for this paper. Please provide another example that shows the difference between ATLASx and the previous algorithms.

Page 18: Conserved atom ratio (CAR) calculated the ratio of the conserved atoms between reactants and products. Is CAR biased to be high for a reactant-product pair that contains large groups (e.g., -CoA)?

Authors response to the reviewers' comments

Title: ATLASx: a computational map for the exploration of biochemical space

Tracking #L: NCOMMS-21-32885

We thank the reviewers for their valuable feedback and for their constructive suggestions. We have been able to incorporate changes to reflect most of the suggested improvements provided by the reviewers. Furthermore, we have improved the user-friendliness of the web service by providing more documentation on how to use the different tools, and we have addressed the issue of open access to our platform.

In the following we provide a point-by-point response to the reviewers' comments and concerns. We have also highlighted the modifications in the revised manuscript.

In addition, as a general remark for all the reviewers, we have prepared a comprehensive comparison between ATLASx and other available tools which highlights the uniqueness of this work. We modified manuscript to highlight the unique utility of ATLASx in the introduction and discussion, and we added the here presented extended review of existing methods to the supplementary information.

The table below provides an overview on popular selected tools and databases that provide reaction prediction, or pathway search:

	Name	Description	Reaction prediction	Pathway prediction	Online accessibility	# of integrated biochemical databases	Scope
Tools	- BNICE.ch - RetroPath - NovoStoic - ReactPRED ...	Retrosynthesis algorithms that use generalized reaction rules to predict metabolic transformations	Yes	Yes	No	1 to 5	Limited to a given research or engineering question
	MetaCyc	Databases of experimentally elucidated metabolic pathways from literature	No	Yes	Yes	1	Limited to the known pathways in a single database
Databases	MINEs	A database that predicts potential biological products for mass-spectrometry applications	Yes	No	Yes	3	Focused on novel structures for mass-spectrometry applications
	enviPath	A database and web server tool for predicting biodegradation mechanisms	Yes	Yes	Yes	Focused on xenobiotic chemicals	Limited to the microbial biotransformation of organic environmental contaminants
	ARBRE	A database and web server tool to biosynthesize aromatic compounds	Yes	Yes	Yes	Focused on aromatic compounds	Limited to the compounds in the proximity of aromatic amino acids
	PathPred	Web based pathway search that uses substrate-product pairs in KEGG database for reaction prediction	Yes	Yes	Yes	1	Limited to KEGG database
	novoPath Finder	Web based pathway search that uses generalized reaction rules	Yes	Yes	Yes	3	Maximum pathway length 10
	- ATLAS of biochemistry, - Transform-MinER	Databases of all possible biochemical reactions among KEGG compounds	Yes	Yes	Yes	1	Limited to KEGG database
	ATLASx	A database of all known and predicted biochemical reactions in the unified space of biological and bioactive compounds	Yes	Yes	Yes	14	Global No limit on the pathway length

Reaction prediction and retrobiosynthesis tools

Diverse computational tools have emerged to bridge knowledge gaps in metabolism through cheminformatic predictions of potential metabolic reactions. Most of these tools have been developed for metabolic engineering applications, where the objective is to find biosynthetic routes that produce a desired target compound in a host organism¹⁻⁵. Identification of these biosynthetic routes is accomplished by biochemically “walking back” from the target to precursor metabolites that are produced by, or fed to, the host organism. This procedure is called *retrobiosynthesis* and is implemented in a range of tools such as BNICE.ch^{6,7}, novoStoic⁸, ReactPRED⁹, and RetroPath^{10,11}. While tools featuring reaction rules have the power to predict novel structures and biochemical

reactions, their application is usually limited to a given research or engineering question. Furthermore, many tools require programming skills and extensive curation, filtration and screening of results. Since these tools use different standards, methods, and biochemical knowledge it is difficult to compare, reproduce, or transfer the results between different research groups.

Database approach based on known reactions

To provide easier access to reaction and pathway prediction for a broader range of researchers various databases and online tools have been developed. The scope, size and application of these databases are different. MetaCyc¹² and KEGG¹³ databases are golden standards to access experimentally observed (known) metabolic pathways. However, many metabolites and bioactive molecules (for example, drug molecules) remain out of scope of KEGG and MetaCyc since these molecules have been identified through mass spectrometry experiments, but their metabolism has not been elucidated yet.

Database approach based on known or novel reactions

To provide more comprehensive resources, the concept of enzymatic rules and reaction prediction is also employed by enviPath¹⁴, a database for predicting biodegradation mechanisms, by MINEs¹⁵, a database that predicts potential biological products for mass-spectrometry applications, and by ARBRE¹⁶ a database centered around industrially important aromatic compounds. All these databases are available as user-friendly web tools. However, the scope of mechanisms integrated in these databases is focused on their specific field of application.

To systematically explore the metabolic dark matter arising from the elasticity of enzymatic catalysis, an unbiased approach is employed by Transform-MinER¹⁷, novoPathFinder¹⁸, and ATLAS of Biochemistry database^{19,20} (developed in our group). These approaches generalize the concept of reaction rule and retrobiosynthesis and predict all theoretically possible novel reactions between metabolites reported in a single or in multiple databases. The generalized network of biochemistry around the KEGG database is published as ATLAS of Biochemistry and Transform-MinER. More recently, novoPathFinder applied this idea in larger scale by integrating the KEGG, ChEBI²¹, and Rhea²² databases. These works on unbiased enzymatic reaction prediction received a lot of attention from the scientific community. For instance, the generated repository of ATLAS allows the user to search for all known or predicted possible routes from any substrate compound to any product in the KEGG database. However, one major drawback of these works is their limitation to a single or few compound sources, which excludes many drugs and plant natural products with undefined or putative biological functions. Predicting enzymatic reactions from biochemical compounds retrieved from other databases will help expand the scope of our predictions and enhance the application range and the predictive power of the database.

ATLASx

Following the previous publications but bringing the concept to the next level, in this article we introduce ATLASx, the first attempt to systematically map and fill the knowledge gaps in metabolism at the scale of the global biochemical knowledge. We decided to expand our database to all known, well-characterized and novel, predicted reactions between millions of known chemicals, biochemical and bioactive compounds. The ATLASx workflow unifies biochemical reactions and compounds from 14 different database sources (4 times bigger than any previous database) into one curated dataset called bioDB. bioDB holds 1.5 million unique biological or bioactive compounds and 56,000 unique

biochemical reactions, which enable the prediction of a hypothetical biochemical space. By applying 490 bidirectional, generalized reaction rules from BNICE.ch onto biological and bioactive compounds within the database, we predicted around 1.6 million potential biotransformations between bioDB compounds. Another 3.6 million reactions were found to connect bioDB compounds with molecules only found in chemical databases, producing a total of 5.2 million predicted reactions. From these predictions, we characterized the connectivity and reactivity of biologically important molecules. The development of ATLASx involves significant advances in database design, data analysis, and programming techniques. The ATLASx platform can be distinguished from the previous works by its comprehensive scope and wide range of applications. ATLASx can be readily used for the design of novel metabolic pathways, and for the exploration and expansion of biosynthesis pathways. Finally, and for the first time, ATLASx provides an estimation on the staggering number of unknowns in biochemistry, and can thus foster future research explorations into metabolic dark matter.

REVIEWER COMMENTS

Reviewer #1 (Expertise: chemical biology):

Hatzimanikatis et al. applied 490 generalized enzymatic reaction rules on 1.5 million biological and bioactive structures to construct a global network of biochemical knowledge map (ATLASx). The known biochemical pathways could be diligently reproduced using ATLASx, however, the capacity of exploring new biosynthetic pathway and unknown biochemical reaction is not well estimated in this work. I agree with the author's view that the exploration of metabolic dark matter arising from the elasticity of enzymatic catalysis is critically important, but I do think that this challenging issue "can be achieved by current computational approaches". I would like to think the ATLAS as a well-established database but not a good predictor after reading the whole paper and visiting their webserver.

My Major Concerns:

1. How can prove the capacity of exploring new biochemical transformation by rule-based method? Are the 490 so-called generalized enzymatic reaction rules sufficient for describing so complicated biocatalysis? They claim that a total of 5.2 million novel reactions were generated by applying the reaction rules. This number is far more than the "valid" reactions (56000 in this work), and it really attracts me if they are truly new. While my queries are, what kind of reaction means novel? how reliable for these hypothetical reactions? is there a score function(not the Gibbs free energy but something like weighted score reflecting the matching between structure and reaction rules) to evaluate the possibility of these reactions? This work is lack of the validation on the virtual bioreaction network, thus the ability of biochemical reaction exploration is not well verified. In sum, they should carefully prove the reliability of computationally constructed map for the hypothetical vicinity of known biochemistry.

Regarding the first request for proof that rule-based methods have the capacity to predict novel, valid biochemical transformations, we already provide a collection of references in the manuscript that illustrate different examples of experimental validation of novel reactions predicted by rule-based methods, including by BNICE.ch (see Introduction section)

Regarding the second question on whether the 489 rules are sufficient to describe the various enzymatic transformations catalyzed by enzymes, we present that out of 56,087 bioDB reactions, 41,680 are considered as “validated”, meaning that they pass our quality criteria as detailed in the manuscript. Out of the validated reactions, our reaction rules can reconstruct their reaction mechanisms in 29,768 cases. Out of these 29,768 reactions, 25,365 are reconstructed by a single reaction rule, while the remaining 3,403 reactions require the application of two or three consecutive reaction rules, ensuring that even more complex reaction mechanisms are reconstructed. These numbers are presented in the Supplementary Table S6, and explained in detail in the Methods section under “Reconstruction of known reactions with BNICE.ch reaction rules”. From this analysis, we obtain a net reconstruction of bioDB reactions of 53.1 %, and a reconstruction of 71.4 % when only considering validated reactions. Given that only 35% of bioDB have a high curation standard (i.e., mass balanced reactions with EC annotation) (for details, see Supplementary Table S4), we consider our coverage of reactions as sufficient. To clarify this point in the manuscript, we added an explanatory sentence to the end of the Methods section on the “Reconstruction of known reactions with BNICE.ch reaction rules”.

Nevertheless, some of the more complex reactions are currently not reconstructed by our rules, for example because their reaction mechanisms are not yet known to science and they are therefore currently out of the scope of our prediction method. One example of such a case is the third reaction step in the reconstruction of the staurosporine biosynthesis pathway: The KEGG reaction R11121 is not reconstructed by any suite of reaction rules in ATLASx, which is due to the fact that the complex reaction mechanism of the responsible enzyme, cytochrome P450 StaP, has not yet been elucidated. We added a short discussion of this case to the section “ATLASx fills metabolic gaps and proposes new biosynthesis pathways” to discuss the point raised by the reviewer. Hence, the fact that such reactions are not covered by our rules makes it possible to detect complex reactions in the biochemical network where experiments are needed to potentially discover new intermediate metabolites or reaction mechanisms.

Curation of the reactions in the public databases is beyond the scope of this work. However, we believe that the preprocessing and classification of the quality of the curation, as performed in our studies, can serve as a starting ground for systematically curating the known databases.

The next point raised by the reviewer concerns the novelty of the predicted reactions: We want to highlight here again that we checked our predicted reactions against all of the known reactions that we could gather from publicly available resources. We claim therefore that the predictions that did not match any of the known, collected reactions are truly novel.

Furthermore, the reviewer questions the reliability of our predictions and suggests the implementation of a scoring function that reflects the probability of the reaction to exist in nature. First, we would like to highlight that our reaction rules are manually constructed based on biochemical textbook knowledge, and that they describe only reaction mechanisms that have been observed in nature. Hence, reactions derived from these transformation rules follow biochemical logic by design. However, we agree with the reviewer that a scoring function that indicates that probability for a novel reaction to exist in nature or to be engineered using protein design techniques would be helpful. For this reason, we provide the BridgIT score on-demand for all of our predicted reactions on our website. As explained in the manuscript, the BridgIT score indicates the similarity of a novel reaction to all known, well-characterized reactions. If BridgIT finds similar reactions (high BridgIT score) within the

space of known reactions, we can assume that the novel reaction in question has a higher probability to be found in nature, or to be engineered.

Other methods also use reaction feasibility estimations to rank predicted reactions. For example, *enviPath*¹⁴ estimates the probability of a predicted reaction to happen in the presence of other possible reactions using machine-learning models trained on biodegradation data. We believe that such feasibility tests based on biochemical prioritization rules, toxicity etc., can be useful to further curate reaction networks around a target biochemical process (see *NICEdrug.ch*²³, *ARBRE*¹⁶).

To address the reviewer's last comment on the validation of the predictions in a network context, we already provide a systematic analysis of the ability of *ATLASx* to reconstruct known biosynthesis pathways from the network of known and predicted reactions (see Results section "Searching for biological pathways within *ATLASx*").

2. According to the main text, when applying the reaction rules to the structures, only the reactions whose products exist in *bioDB* or *chemDB* were kept. Will novel structures be produced in this process? If yes, I think the new structures are also important to expand the chemical space of biological reactions. So additional database containing the reactions with "virtual" components could be constructed to supplement *bioATLAS* and *chemATLAS*.

We agree with the reviewer that the novel structures produced in the process of reaction prediction are important to further expand the chemical space. However, in this study we focus on feasible molecular structures that are observed and cataloged in databases.

Integrating all novel structures into the *ATLASx* database would increase the size of the network by orders of magnitudes and is beyond the scope of this work. Nevertheless, and as discussed in the Discussion part of our manuscript, targeted integration of novel structures that are relevant to bridge knowledge gaps in the global biochemical transformation network is highly relevant and will be the topic of future research.

3. I'm confused about the classification of biological and chemical compounds (i.e. the *bioDB* and *chemDB*). Since this work aims to explore the metabolic "dark matter", why not just classified compounds as biological origin (or metabolites) and non-biological origin (including those unlabeled from *PubChem* and other data sources). For examples as summarized in the supplementary Table S1, I do not understand why put bioactive structures into *bioDB*. Why put the metabolic data from *MetaNetX* into bioactive subset of *BioDB*? There are too many details for the construction of database and reaction network in the main text, this manuscript is suitable for professional reader while not general and friendly for diverse readership.

We thank the reviewer for pointing out that our classification of compounds and its justification are not clear to the reader. The first part of the answer is the that the division into biological, bioactive and chemical compounds is artificial in the sense that that the classification derives from fields of research rather than the natural origin of compounds. As a matter of fact, the true origin of a compound is not always clear and unique. For example, a secondary plant metabolite might at the same time be a product of chemical synthesis, and molecules used in organic chemistry and labeled as synthetic may occur naturally in petroleum that was formed by natural processes millions of years ago. By taking the database perspective, we give less weight to the true origin of a molecules, and we consider their level of annotation instead. In other words, we care about how much is known about a

molecule in terms of biochemistry. Does it have a known role in metabolism (biological)? Has it been observed to interfere with biological systems (bioactive)? Or was it just detected in a mass spectrometry experiment looking for new secondary metabolites, and is not further situated in a biological context (chemical)? To give another example, biological/bioactive databases like KEGG and MetaCyc provide degradation pathways for xenobiotic molecule of synthetic origin, and the chemical database is the most comprehensive source of secondary metabolites since it stores any molecule ever found in nature (e.g., structure discovered by mass spectrometry screening) without requiring further curation standards or functional annotation. These examples show how difficult it is to accurately devise the chemical compound space according to the origin of the molecular structures. In contrast, the distinction of molecules by databases generally reflects the level of biochemical knowledge available for a given compound, which is also more relevant for our study since we aim to integrate compounds into the global reaction network that were previously not reported to interact with biochemical processes. We therefore decided to leverage the database attribution to classify compounds by their level of available biochemical knowledge.

To clarify this point in the manuscript, we elaborated on the classification of compounds in the manuscript (see Results section, “bioDB unifies over 1.5 million unique compounds”).

4. For the webservice, users could not use the web tool listed in the manuscript, if without registration. And the registration for academic use seems cumbersome that too many personal and organization information are required to finally get the License Agreement. In my experience, the software or database are free for academic use and the interface are very friendly for many computational paper published in nature serial journal. While it is so hard for result reproduction by visiting their webservice.

Regarding the availability of ATLASx, the use of the website requires a simple registration, which is free and without discrimination to any entities or persons for non-commercial uses. It has been introduced to prevent abuses of our computational resources (and also use of web robots) to maintain the integrity and quality of our website, so that it can better serve the research community. This set-up also complies with certain legal restrictions that are imposed on the use of the data.

We addressed the issue of transparency of the methodology by adding several elements to the main manuscript, the Methods part, the supplementary material, the web server and the Git repository. To improve the reproducibility of our analysis, we added the code used to perform the pathway reconstruction analysis and the network extraction to a public git repository available at <https://github.com/EPFL-LCSB/ATLASxAnalyses>. The code is well documented and contains instructions to reproduce the results presented in the manuscript. The link to the git repository has been added to the resubmitted version of the manuscript. We further addressed the reviewer’s questions on methodology by clarifying the explications in the manuscript and the methods section, and we added supplementary information where we thought it would help to improve the overall transparency of our approach. Finally, we facilitate the reproducibility of our analyses through improved documentation on how to use our online tools.

As a final remark to the first reviewer’s general comment, we would like to mention that we disagree with the reviewer on their last point in the summary statement: “I would like to think the ATLAS as a well-established database but *not a good predictor* after reading the whole paper and visiting their webservice”. ATLASx as well its previous edition, ATLAS, have been successfully harnessed by others to design new metabolic circuits (e.g., starch synthesis from CO₂^{24,25}) and pathways (e.g., tropane

derivatives ²⁶). These examples clearly demonstrate the utility of our databases for predicting and designing metabolism.

Reviewer #2 (Expertise: metabolic engineering):

In this paper, MohammadiPeyani et al. expand our current knowledge of the metabolic “dark matter,” the set of possible reactions extrapolated from the known metabolome using a set of generalized enzymatic reaction rules. The authors first generate a unified biochemical database and a chemical one. Using a previous list of 490 generalized enzymatic reactions rules obtained from their BNICE tool, they generate a new set of hypothetical biochemical reactions. While the methodology is similar to their previous works (Hadadi et al. 2016, Hafner et al. 2020), the novelty comes from the starting compound database used. In this study, >5M novel reactions could be inferred versus 150’000 from the previous version (Hafner et al. 2020). ATLASx will be a much-needed tool to find novel pathways for metabolic engineering and should be of interest to the readership of Nature Communications. The study is well-written with descriptive figures, but additional information could be added to the manuscript to ensure reproducibility and clarity.

Specific points that could be addressed to improve the manuscript:

Regarding the accessibility of ATLASx, I would like to congratulate the authors for making ATLASx available through an online platform. From a user perspective however, accessibility to the database needs to be applied for and approved through the principal investigator. Since the database would probably be mostly used by PhD students and postdocs, I would suggest giving direct access to all researchers regardless of their academic position.

We would like to thank reviewer for their feedback. Currently, we do provide access to PhD students and postdocs, but we verify their affiliation to academic institutions. Basically, the use of the website requires a simple registration, which is free and without discrimination to any entities or persons for non-commercial uses. It has been introduced to prevent abuses of our computational resources (and also use of web robots) to maintain the integrity and quality of our website, so that it can better serve the research community. This set-up also complies with certain legal restrictions that are imposed on the use of the data.

Central to this study are the 490 reaction rules used from BNICE. However, details regarding these rules were absent from the main and supplementary text. Were they derived from KEGG as in the previous studies? Looking at the KEGG database, the current release contains ≈12’000 reactions (≈11’000 reactions in 2018), is the >20% increase in reaction rules (400 in 2018 vs 490 now) only due to the addition of 1’000 new reactions?

We thank the reviewer for this comment and we agree that this crucial information was previously missing from the manuscript. To amend this, we added the Supplementary Table S8 to show the evolution of our set of rules from 2015 to 2018 to 2020 (present work).

First of all, we would like to point out that the exact number of rules is 489, and not 490 as reported in the initial submission of the manuscript. This mistake is due to careless rounding on our side, and we now changed the manuscript to always state the exact number.

The 89 new rules focus on reaction mechanisms from plant and bacterial secondary metabolism, and also on mechanisms involved in the degradation of environmental contaminants. We further added 5 non-enzymatic reaction rules to include reactions that occur spontaneously under biological conditions. Out of the 89 new rules, only three introduced a new third-level EC number to our collection, while the big majority of new rules contributed a new reaction mechanism to an existing third-level EC number (e.g., introduction of new cofactor, slight modification in the structure surrounding the reactive site of a substrate).

Hence, to answer the question about KEGG, there is no direct link between new rules and new reactions in KEGG. First of all, most of the newly introduced reactions in KEGG are already covered by existing reaction rules. Second, the new mechanisms were added because we intensively studied specific parts of metabolism, such as the biosynthesis of secondary plant metabolites or the degradation of organic pollutants. Diving into these less common biochemistries lead to a refinement of rules that were relevant for the given topics.

To clarify this matter, we added explanatory text to the Methods section, including the reference to the Supplementary Table S8.

p.7 The authors mentioned different curation standards across the different databases used. Does the unified version contain all reactions from these databases, others than the duplicate reactions that got removed? For instance, were unbalanced reactions kept in the analysis? Can the authors comment on whether filtering criteria were applied here and the rationale behind the inclusion or occlusion of those?

We have collected, unified and curated the information of compounds and reactions from external sources.

Compound integration:

Compound structures and properties were collected from KEGG ¹³, SEED ²⁷, HMDB ²⁸, MetaCyc ¹², MetaNetX ²⁹, DrugBank ³⁰, ChEBI ²¹, ChEMBL ³¹, BRENDA ³², Rhea ²², BiGG models ³³, Reactome ³⁴, and BKMS-react ³⁵. We then unified the space of collected structures based on their canonical SMILES generated with the OpenBabel library. The compounds were filtered to keep only those that had a defined, single molecular structure. Compounds that did not fit this criterion (i.e., salts, polymers, generic molecules with undefined branches) were excluded.

Reaction integration:

Reactions were unified based on the structure of their reactants and products (as explained above). In terms of filtration, transport and stereoisomeric reactions were filtered out to only keep reactions that modify the connectivity of atoms within the molecule. During the curation of imported reactions, we annotated them with information on atom-balance, BNICE.ch rules and, estimated Gibbs free energy (based on group contribution method) if possible.

Therefore, unbalanced were kept in database including all available identifiers from different databases and all their properties calculated or reported. However, one can avoid such poorly defined reactions that are not balanced and/or orphan by only retaining reactions with associated BNICE.ch rules in the ATLASx pathway search/network extraction.

p. 11, l. 3 The top 100 pathways relative to what metric?

p. 11 “Out of the reconstructed pathways, 40% were exactly reconstructed within the top 100 pathways” This sounds low but again what is meant by “top 100” pathways?

Since a similar point was raised by another reviewer, we realized that it will be better to present the comparison with the MetaCyc pathways in a different manner. In the previous version of the manuscript, we applied the pathways search algorithm NICEpath, which finds the shortest pathways according to the distance calculated based on atom conservation within reactant-product pairs (Conserved Atom Ratio, CAR ³⁶). We restricted the maximum number of pathways to 100, therefore restricting our prediction space to the best 100 pathways according to the overall atom conservation along the pathway (top 100 pathways). In the new version of the analysis, instead of restricting the maximum number of pathways we decided to determine the rank of the native MetaCyc pathway within the output of the ATLASx pathway search. The results of this analysis were added to the manuscript, replacing the previous analysis. Also, the corresponding parts of the Methods and Results have been rewritten (“Searching for biological pathways within ATLASx”), and the Figure 3 has been changed. In the current version of the manuscript the top 100 pathways are not mentioned anymore. Instead, the readers can see the exact rank of the pathway within the output of the pathway search according to the NICEpath algorithm. For the native pathways that were down in the ranking list, we provide an explanation in the manuscript and an example (Figure 3c).

Minor comments:

p. 4 “Yang et al. were able to experimentally validate predicted ATLAS reactions” reference missing in text

Thank you for this comment. We added the missing reference to the manuscript.

ref. 57 authors name in capital letters

Thank you for this comment. We corrected the reference in the manuscript.

Reviewer #3 (Expertise: Metabolic engineering, industrial, retrobiosynthesis):

Authors implemented BNICE.ch to construct a biochemical knowledgebase, ATLASx, which guides the analysis of unknown metabolic processes. Combined with the enzyme prediction tool, BridgIT, it is expected that the predicted biosynthetic pathways will help developing microbial cell factories more efficiently. The followings need to be considered to improve the manuscript.

Page 5: The authors implemented 490 bidirectional reaction rules from BNICE.ch. However,

unidirectional reactions are also prevalent and important for the construction of reaction networks. If applicable, please add unidirectional reaction rules.

Generalized enzymatic reaction rules reflect the structural properties of an enzyme's active site, but they do not include information about the directionality of the reaction catalyzed by the enzyme. This is due to the fact that the directionality of a reaction is a thermodynamic property and depends on the concentration of substrates and products (in the majority of cases). Therefore, the operating Gibbs free energy of a reaction determines its directionality of the reaction while preserving the notion of the catalytic reversibility of the enzyme. The reversibility of enzymatic reactions is the first reason why all of the reaction rules in BNICE.ch are bidirectional. The second, more important reason is that BNICE.ch is designed to both predict potential products from substrates, and also potential substrates from products. Reversing the direction of reasoning in reaction prediction is particularly important for retrobiosynthesis applications, where one biochemically walks back from a target chemical to find biochemical routes and metabolite precursors that allow for the biosynthesis of the target chemical. For this application, bidirectional reaction rules are necessary, including rules that represent irreversible enzymatic processes in a bidirectional manner. In the specific case of ATLASx construction, bidirectional rules allow us to not only predict which chemDB compounds can be derived from bioDB compounds, but also which bioDB compound may be derived from a chemDB compound. A practical application of this would be the prediction of potential biological biodegradation products (in bioDB) for xenobiotic chemicals (in chemDB). In the case where reaction directionality needs to be constrained, further analyses involving thermodynamic flux analysis (TFA, beyond the scope of this work) can be performed in the context of a metabolic model.

Page 5: bioDB unified 1.5 million unique compounds in 2D structural compound entries. Can the entries show differences between isomers? Does the ATLASx contain compounds with various isomers and corresponding reactions?

We assume the reviewer pointed to the following sentence in the manuscript:

“As a result of the unification procedure, a unique compound entry in bioDB can contain different resonance forms, stereoisomers, as well as dissociated and charged states of a same compound “.

Within ATLASx the stereoisomers of a compound are merged into one entry. To achieve proper generalization of the reaction mechanisms in the reaction rules, our rules can process all the stereoisomer forms at the same time. Therefore, for BNICE.ch the stereoisomers of a same compound are indistinguishable, and so they are for the reactions in the ATLASx database. For reaction and pathway prediction we consider that it is important to focus the on the changes in molecular structure and atom composition of the compounds, and to not restrict the search to specific stereoisomers. We suggest that stereochemical considerations are to be performed after the reaction and pathway predictions are found. Converting the 2D description of the compounds into all possible 3D stereoisomeric forms can be performed using cheminformatic software.

To address the stereochemistry considerations and we added the following part to the Methods part of the manuscript:

“Since ATLASx does not distinguish between stereoisomers, users who wish to include stereochemical considerations are encouraged to post-process ATLASx output with external cheminformatic software (e.g., `rdkit.Chem.EnumerateStereoisomers` module in RDKit) to expand the 2D structures to all possible 3D stereoisomers structures.”

Page 6: The compiled compounds drugs (or drug-like compounds) broaden the utility of the database for drug discovery. Then, it would be beneficial to provide molecular property distribution of the compiled compounds (e.g., partition coefficient, quantitative estimate of drug-likeness).

We thank the reviewer for this comment. It is possible to calculate further molecular properties but it needs to be applied in higher level and more focused databases. Recently, we have published an online drug discovery platform, called NICEdrug.ch ²³, which exclusively focuses on drugs and drug-like compounds. In NICEdrug.ch, we evaluate compounds in terms of toxicity, metabolic fate, and drug repositioning. Further analysis on drug molecules remains out of scope of the current study since it is discussed in another publication. It is important to highlight that the compounds in ATLASx and NICEdrug.ch are catalogued with the same identifiers, and therefore data can be easily transferred between these platforms on our website.

Furthermore, all the compounds in ATLASx have structural information and they are linked to at least one external source where specific molecular properties can be queried or calculated.

Page 8: Almost all molecules in bioDB were applicable for the BNICE.ch reaction rules. Was it due to the specificity of the reaction rules for molecules in bioDB? Can molecules not in the bioDB have the potential to undergo biochemical transformations?

We thank the reviewer for these interesting questions. Regarding the first question, it seems logical that the specificity of reaction rules to biological molecules favors the recognition of both biological and bioactive molecules. However, many bioactive molecules act as inhibitors, and they are not necessarily involved in metabolic reactions. For example, a molecule can interact with a protein receptor, or it can block an active site of an enzyme without undergoing catalysis. Therefore, our reaction rules, which are based on metabolic reactions, are not designed specifically for all bioDB molecules, yet can recognize a majority of bioDB compounds.

Regarding the second question, the generalized enzyme reaction rules provide information about the reactive site, i.e., the substructure on the substrates that is critical for binding to the protein pocket and performing reaction. Therefore, reaction rules can identify reactive sites on any compounds that harbors the pre-defined substructure describing the reactive site. Since the rules are bidirectional, we can assume that all chemDB compounds that were derived from bioDB compounds in ATLASx (863,000) have at least one reactive site, meaning these chemically originated compounds are situated only one reaction step away from a known biological or bioactive compound and they can be integrated in biochemical networks. In the future, we will also screen all chemDB compounds for biochemical reactive sites and integrate them into potential biochemical reactions. In a preliminary reactive site analysis, we ranked all PubChem compounds based on their molecular weight. Starting from lightest ones, we analysed 5 million compounds. In this set, 56,000 out of 5 million PubChem molecules have been found to contain at least one reactive site. Comparing to biological compounds, we therefore estimate that the biological activity of PubChem compounds to be orders of magnitude lower.

Page 10: Please elaborate on the general description of the NICEpath (how it works, what does the ranking means.) in the methods section.

The general description of the NICEpath in the Methods section has been modified to better explain the underlying concept of the pathway search. We also refer the interested reader to the NICEpath publication for a more detailed description of the method.

Page 11: The authors searched the top 100 pathways to validate the pathway search algorithm. However, 100 predictions are still too much to examine in wet experiments. Performance evaluation of general retrosynthesis algorithms analyze with top-1, top-5, top-10 searches. Perform the analysis with various thresholds.

We repeated the analysis with the different thresholds. To do this, we found the rank of each of the native MetaCyc pathways in the ATLASx network. The rank ranges we analyzed were: '1', '2', '3', '4', '5', '6-10', '10-15', '16-20', '21-50', '51-100', '101-300', '301-500', '501-1000', '1001-5000', '5001-10000', '10000+' (Updated Figure 3b of the manuscript). We allowed pathways with low ranks because the standard pipeline for metabolic pathway design that we envision includes several steps after the pathway search that are aimed to ensure that the proposed pathways are viable within the host organism and produce the highest possible yield. We extensively elaborated on the steps of pathway design in the "Pathway design" chapter of the book *Metabolic Engineering: Concepts and Applications* (Sang Yup Lee (Editor), Jens Nielsen (Editor), Gregory Stephanopoulos (Editor), ISBN: 978-3-527-34662-2, June 2021). There we divided the pathway design into 4 stages: (i) creation of the biochemical network; (ii) pathway search; (iii) enzyme assignment; (iv) pathway evaluation in the chassis model. The results we provide correspond to the stages (i) and (ii) of the pathway design pipeline, therefore, we allow any number of pathways to be generated as the number will be reduced for the experimental validation in the steps (iii) and (iv) of the general pipeline. The BridgIT predictions which we provide as integrated functionality of ATLASx allow users to further narrow down the list of pathways based on enzyme availability (step iii). We also provide pathway export in as a .tsv file, which contains all the compound information and reaction stoichiometry that is necessary to integrate the predicted pathways into a stoichiometric model of the host organism and to evaluate them using standard libraries, such as COBRA and TFA, for flux balance and thermodynamic flux analysis (step iv). Regarding the pathway search algorithm validation, we validated the algorithm on the dataset of MetaCyc reactions and we observed that 85% of the pathways were within top-15. We argue that the pathways that we do not reconstruct are especially susceptible to shortcuts as they have steps where the atom conservation can be improved. We provide an example of such a case (Figure 3c). Since many of the pathways in the MetaCyc pathway dataset were branching, we linearized them for comparability with our results. Potential pitfalls of such linearization are addressed in Supplementary table S7. The simultaneous reconstruction of all the branches required for biosynthesis of a compound lies beyond the scope of the current work and will be addressed in our future research.

Page 11: The authors showed 99% of the precursor-target pairs were found. As the former analysis performed the predictions with the threshold of top 100 pathways, the value would be better to be 46% (which was the number of exactly reconstructed pathways within the top 100 pathways). Otherwise, it might confuse readers.

As we decided to remove the top 100 pathways threshold, we consider all the pathways that are present in the network to be reconstructed (99% of the pathways). To avoid confusion, we removed

any mention of the top-100 pathways from the manuscript. Moreover, we provide the detailed ranking for the 99% of the reconstructed pathways (Fig. 3b).

Page 13: The investigation of the biosynthesis pathway of staurosporine is the application of the previously developed algorithms (BNICE.ch and BridgIT), rather than the novel analysis for this paper. Please provide another example that shows the difference between ATLASx and the previous algorithms.

While such a pathway expansion could have been performed in a similar way using previously developed algorithms, we would like to point out that the strength of ATLASx lies in its pre-compiled networks of predicted reactions, its database and its web interface. ATLASx for the first time allows this kind of investigation to (i) be performed online, thanks to the ATLASx website, (ii) in a short time, because the pre-existing network of predicted reactions allows for a rapid extraction of the requested subnetwork, and (iii) to be reach as many reaction steps away from the original compound or pathway as demanded by the user. The last point in particular would not be feasible using BNICE.ch only, because of the exponential nature of iterative, rule-based network generation.

Hence, the same analysis was not only previously impossible to perform in this way, it also has been made accessible to scientists for exploration thanks to the pre-compiled networks of predicted reactions, and thanks to the ATLASx web interface.

Page 18: Conserved atom ratio (CAR) calculated the ratio of the conserved atoms between reactants and products. Is CAR biased to be high for a reactant-product pair that contains large groups (e.g., -CoA)?

Yes, the CAR corrects for large groups by replacing the CoA moiety with a “CoA atom” in the case that CoA only appears in either the substrate or the product of the reaction. For more details on the CAR calculation, we refer the reader to the NICEpath publication³⁶ in the manuscript.

References

1. Bachmann, B. O. Biosynthesis: Is it time to go retro? *Nature Chemical Biology* **6**, 390–393 (2010).
2. Hadadi, N. & Hatzimanikatis, V. Design of computational retrobiosynthesis tools for the design of de novo synthetic pathways. *Current Opinion in Chemical Biology* **28**, 99–104 (2015).
3. Wang, L., Ng, C. Y., Dash, S. & Maranas, C. D. Exploring the combinatorial space of complete pathways to chemicals. *Biochemical Society transactions* **46**, 513–522 (2018).
4. Lin, G.-M. M., Warden-Rothman, R. & Voigt, C. A. Retrosynthetic design of metabolic pathways to chemicals not found in nature. *Current Opinion in Systems Biology* **14**, 82–107 (2019).
5. Jeffryes, J. G., Seaver, S. M. D., Faria, J. P. & Henry, C. S. A pathway for every product? Tools to discover and design plant metabolism. *Plant Science* **273**, 61–70 (2018).
6. Hatzimanikatis, V. *et al.* Exploring the diversity of complex metabolic networks. *Bioinformatics* **21**, 1603–1609 (2005).
7. Tokic, M. *et al.* Discovery and Evaluation of Biosynthetic Pathways for the Production of Five Methyl Ethyl Ketone Precursors. *ACS Synthetic Biology* **7**, 1858–1873 (2018).
8. Kumar, A., Wang, L., Ng, C. Y. & Maranas, C. D. Pathway design using de novo steps through uncharted biochemical spaces. *Nature Communications* **9**, 184 (2018).
9. Sivakumar, T. V., Giri, V., Park, J. H., Kim, T. Y. & Bhaduri, A. ReactPRED: a tool to predict and analyze biochemical reactions. *Bioinformatics* **32**, 3522–3524 (2016).

10. Delépine, B., Duigou, T., Carbonell, P. & Faulon, J.-L. RetroPath2.0: A retrosynthesis workflow for metabolic engineers. *Metabolic Engineering* **45**, 158–170 (2018).
11. Koch, M., Duigou, T. & Faulon, J.-L. Reinforcement Learning for Bioretrosynthesis. *ACS Synth. Biol.* **9**, 157–168 (2020).
12. Caspi, R. *et al.* The MetaCyc database of metabolic pathways and enzymes. *Nucleic Acids Research* **46**, D633–D639 (2018).
13. Kanehisa, M. & Goto, S. KEGG: kyoto encyclopedia of genes and genomes. *Nucleic Acids Research* **28**, 27–30 (2000).
14. Wicker, J. *et al.* enviPath – The environmental contaminant biotransformation pathway resource. *Nucleic Acids Res* **44**, D502–D508 (2016).
15. Jeffryes, J. G. *et al.* MINEs: open access databases of computationally predicted enzyme promiscuity products for untargeted metabolomics. *Journal of Cheminformatics* **7**, 44 (2015).
16. Sveshnikova, A., MohammadiPeyhani, H. & Hatzimanikatis, V. ARBRE: Computational resource to predict pathways towards industrially important aromatic compounds. 2021.12.06.471405 <https://www.biorxiv.org/content/10.1101/2021.12.06.471405v1> (2021) doi:10.1101/2021.12.06.471405.
17. Tyzack, J. D., Ribeiro, A. J. M., Borkakoti, N. & Thornton, J. M. Exploring Chemical Biosynthetic Design Space with Transform-MinER. *ACS Synth. Biol.* **8**, 2494–2506 (2019).
18. Ding, S. *et al.* novoPathFinder: a webserver of designing novel-pathway with integrating GEM-model. *Nucleic Acids Res* **48**, W477–W487 (2020).
19. Hadadi, N., Hafner, J., Shajkofci, A., Zisaki, A. & Hatzimanikatis, V. ATLAS of Biochemistry: A Repository of All Possible Biochemical Reactions for Synthetic Biology and Metabolic Engineering Studies. *ACS Synthetic Biology* **5**, 1155–1166 (2016).
20. Hafner, J., MohammadiPeyhani, H., Sveshnikova, A., Scheidegger, A. & Hatzimanikatis, V. Updated ATLAS of Biochemistry with New Metabolites and Improved Enzyme Prediction Power. *ACS Synthetic Biology* **9**, 1479–1482 (2020).
21. Hastings, J. *et al.* ChEBI in 2016: Improved services and an expanding collection of metabolites. *Nucleic Acids Research* **44**, D1214 (2016).
22. Morgat, A. *et al.* Updates in Rhea—a manually curated resource of biochemical reactions. *Nucleic acids research* **43**, D459–64 (2015).
23. MohammadiPeyhani, H. *et al.* NICEdrug.ch, a workflow for rational drug design and systems-level analysis of drug metabolism. *eLife* **10**, e65543 (2021).
24. Yang, X. *et al.* Systematic design and in vitro validation of novel one-carbon assimilation pathways. *Metabolic Engineering* **56**, 142–153 (2019).
25. Cai, T. *et al.* Cell-free chemoenzymatic starch synthesis from carbon dioxide. *Science* **373**, 1523–1527 (2021).
26. Srinivasan, P. & Smolke, C. D. Biosynthesis of medicinal tropane alkaloids in yeast. *Nature* **2020** 585:7826 **585**, 614–619 (2020).
27. Overbeek, R. *et al.* The Subsystems Approach to Genome Annotation and its Use in the Project to Annotate 1000 Genomes. *Nucleic Acids Research* **33**, 5691–5702 (2005).
28. Wishart, D. S. *et al.* HMDB: the Human Metabolome Database. *Nucleic acids research* **35**, D521–6 (2007).
29. Moretti, S. *et al.* MetaNetX/MNXref – reconciliation of metabolites and biochemical reactions to bring together genome-scale metabolic networks. *Nucleic Acids Research* **44**, D523–D526 (2016).
30. Wishart, D. S. *et al.* DrugBank 5.0: A major update to the DrugBank database for 2018. *Nucleic Acids Research* **46**, D1074–D1082 (2018).
31. Gaulton, A. *et al.* The ChEMBL database in 2017. *Nucleic Acids Research* **45**, D945–D954 (2017).
32. Schomburg, I. *et al.* BRENDA: a resource for enzyme data and metabolic information. *Trends in Biochemical Sciences* **27**, 54–56 (2002).

33. Schellenberger, J., Park, J. O., Conrad, T. M. & Palsson, B. Ø. BiGG: a Biochemical Genetic and Genomic knowledgebase of large scale metabolic reconstructions. *BMC Bioinformatics* **11**, 213 (2010).
34. Croft, D. *et al.* Reactome: a database of reactions, pathways and biological processes. *Nucleic Acids Research* **39**, D691–D697 (2011).
35. Jeske, L., Placzek, S., Schomburg, I., Chang, A. & Schomburg, D. BRENDA in 2019: a European ELIXIR core data resource. *Nucleic Acids Research* **47**, D542–D549 (2019).
36. Hafner, J. & Hatzimanikatis, V. NICEpath: Finding metabolic pathways in large networks through atom-conserving substrate–product pairs. *Bioinformatics* **37**, 3560–3568 (2021).

Reviewers' Comments:

Reviewer #1:

Remarks to the Author:

The authors have updated their manuscript to reflect the most comments given previously, and manuscript is acceptable now but I still have two minor concerns.

1. For the validity of the reactions, the evaluation by BridgIT scores is acceptable to me. But I am still confused for that only 29,768 out of 41,680 can be reconstructed by their reaction rules, which means the reconstruction rate is not high. Since the reactions were fully based on and predicted by the rules, why so many reactions cannot be reconstructed?

2. The author claimed in their response that "reach as many reaction steps away from the original compound or pathway" is the most important point compare to BNICE.ch, while in the current work, the exploration was performed only once (one round?). Dose this mean that only one step away from the original compound can be explored. How about exploring a next round using the predicted compounds? This might be beyond the scope of the current work, but a discussion about its feasibility, time spent and application would be helpful for users.

Reviewer #2:

Remarks to the Author:

The authors have now addressed most of my concerns and the manuscript is improved with the additional data.

Reviewer #3:

Remarks to the Author:

Authors revised their manuscript appropriately. I do not have any further comment. A very nice work !

Authors response to the reviewers' comments

Title: ATLASx: a computational map for the exploration of biochemical space

Tracking #L: NCOMMS-21-32885

We thank the reviewers for their positive feedback and the editor for the acceptance of our manuscript. In the following, we provide answers to the first reviewer's questions.

REVIEWER COMMENTS

Reviewer #1 (Expertise: chemical biology):

The authors have updated their manuscript to reflect the most comments given previously, and manuscript is acceptable now but I still have two minor concerns.

1. For the validity of the reactions, the evaluation by BridgIT scores is acceptable to me. But I am still confused for that only 29,768 out of 41,680 can be reconstructed by their reaction rules, which means the reconstruction rate is not high. Since the reactions were fully based on and predicted by the rules, why so many reactions cannot be reconstructed?

There are two ways of interpreting a reaction reconstruction of 71%: Either the rules are not sufficient to explain 29% of reactions mechanism, or the reactions provided by external database are not up to our quality standards in terms of reaction mechanism annotation. While the reality might lie somewhere between the two extremes, we give more importance to the second interpretation for the following reason: BNICE.ch rules can reconstruct 29,140 reactions, while only 19,905 of collected reactions are both balanced and have EC class annotated. ATLASx therefore assigns 3 level EC class to 9,235 reactions for which a mechanistic reaction annotation is lacking. Reactions for which a mechanistic annotation is available but that are not covered by BNICE.ch rules are usually difficult to generalize (e.g., the reaction mechanism of the enzyme lanosterol synthase in the biosynthesis of sterols). These reactions, however, are still included in ATLASx and are considered in the pathway search, although their potential for generalization is too low to construct a generalized reaction rule. Therefore, we claim that the number of the reactions that cannot be reconstructed with generalized reaction rules (11,912 reactions) does not illustrate a gap in the current rules, but rather the flaws of the biochemical annotation of the reactions that exist in the current open biochemical databases, or reactions without potential for generalization. The following paragraph to the manuscript is clarifying the point: "Considering that only 35% of bioDB have a high curation standard (i.e., mass balanced reactions with EC annotation) (Supplementary Table 4), we consider our coverage of known reactions as sufficient, and we show that we can propose reaction mechanisms for known reactions for which the reaction mechanism is still unknown."

2. The author claimed in their response that "reach as many reaction steps away from the original compound or pathway" is the most important point compare to BNICE.ch, while in the current work, the exploration was performed only once (one round?). Dose this mean that only one step away from the original compound can be explored. How about exploring a next round using the predicted

compounds? This might be beyond the scope of the current work, but a discussion about its feasibility, time spent and application would be helpful for users.

It is correct that the exploration was performed one reaction step away from the known metabolic space. However, this does not mean that “only one step away from the original compound can be explored”: The exploration around a compound (or a pathways) can be performed for as many steps as required by the user. The user might therefore find derivatives of the original compound that belong to the chemical space, but they will all be connected to metabolites from the biological space within one reaction step.

As the reviewer pointed out, expanding the ATLASx network further into the chemical space, and even the novel compound space, beyond the scope of this work. Such expansion requires a much bigger research effort and it is the subject of ongoing research in our lab.